# SemiReward: A General Reward Model for Semi-supervised Learning

**Siyuan Li**[1,2*] **Weiyang Jin**[2*] **Zedong Wang**[2] **Fang Wu**[2] **Zicheng Liu**[1,2] **Cheng Tan**[1,2] **Stan Z. Li**[2†]

AI Lab, Research Center for Industries of the Future, Hangzhou, China;

[1]Zhejiang University, College of Computer Science and Technology;    [2]Westlake University

`lisiyuan@westlake.edu.cn; wayneyjin@gmail.com;`
`{wangzedong; wufang; liuzicheng; tancheng; stan.zq.li}@westlake.edu.cn`

## Abstract

Semi-supervised learning (SSL) has witnessed great progress with various improvements in the self-training framework with pseudo labeling. The main challenge is how to distinguish high-quality pseudo labels against the confirmation bias. However, existing pseudo-label selection strategies are limited to pre-defined schemes or complex hand-crafted policies specially designed for classification, failing to achieve high-quality labels, fast convergence, and task versatility simultaneously. To these ends, we propose a **Semi**-supervised **Reward** framework (**SemiReward**) that predicts reward scores to evaluate and filter out high-quality pseudo labels, which is pluggable to mainstream SSL methods in wide task types and scenarios. To mitigate confirmation bias, SemiReward is trained online in two stages with a generator model and subsampling strategy. With classification and regression tasks on 13 standard SSL benchmarks across three modalities, extensive experiments verify that SemiReward achieves significant performance gains and faster convergence speeds upon Pseudo Label, FlexMatch, and Free/-SoftMatch. Code and models are available at `https://github.com/Westlake-AI/SemiReward`.

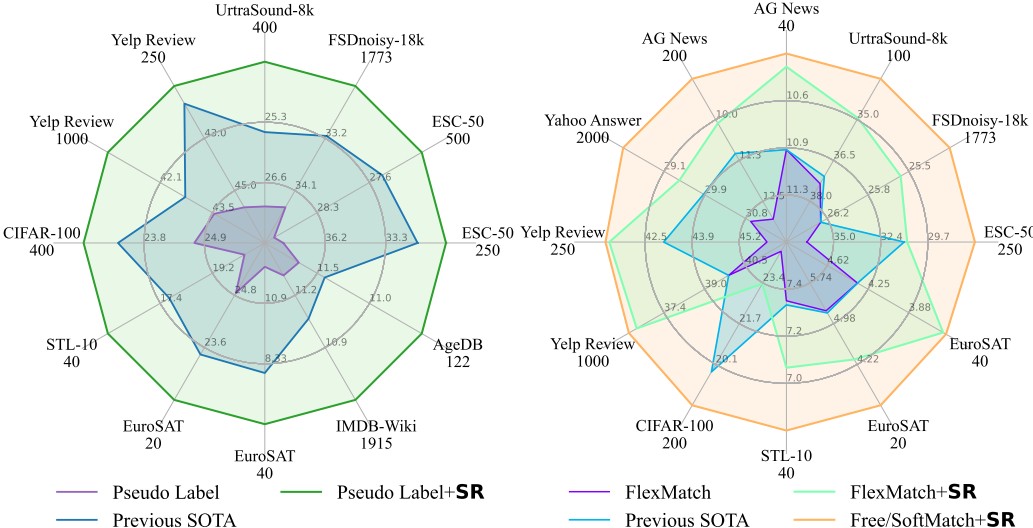

(a) SSL classification and regression benchmarks   (b) SSL classification benchmarks with SOTA methods

Figure 1: **SemiReward** (abbreviated as **SR**) enables existing SSL methods to select high-quality pseudo labels on both classification and regression tasks with fast convergence speeds (Figure 2). Error rates of SSL algorithms are plotted on CV, NLP, and Audio datasets. Note that **previous SOTA** marks the best performance among a set of methods, which denotes 4 general SSL methods used for classification and regression tasks in (a) and 17 SSL methods in USB (Wang et al., 2022a) in (b). SemiReward noticeably improves performance when plugged into existing SSL methods.

---

*First two authors contribute equally.   †Corresponding author.

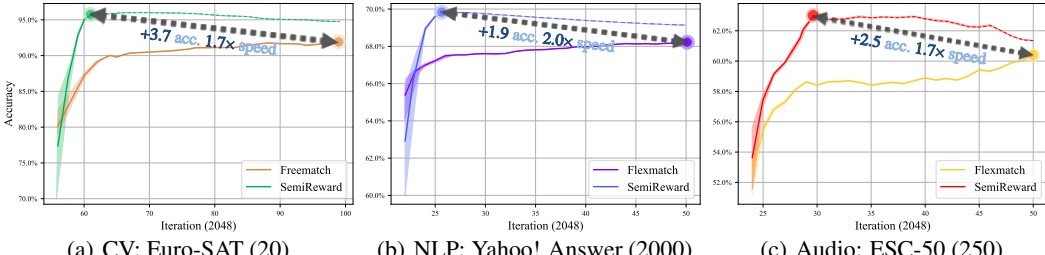

(a) CV: Euro-SAT (20)  (b) NLP: Yahoo! Answer (2000)  (c) Audio: ESC-50 (250)

Figure 2: Top-1 accuracy *v.s.* training iterations ($\times 2048$) on SSL datasets (the number of used labels) of three modalities. Employing **SemiReward** with SOTA SSL methods produces $+1.9\sim3.7$ performance gains with at least 1.7 times fewer training iterations compared to the baseline. We apply early-stop when the validation performance reaches the peak.

## 1 INTRODUCTION

In the past decades, deep learning (DL) has made great progress in various applications with different modalities (He et al., 2016; Devlin et al., 2018; Dong et al., 2018; Li et al., 2024). However, most tasks are in a supervised learning (SL) manner that requires manually labeling data, which is limited in quantity and labor-exhaustive. To extend SL with massive unlabeled data, semi-supervised learning (SSL) exploits the information of unlabeled data with limited labeled data (Tarvainen & Valpola, 2017; Sohn et al., 2020) in the self-training paradigm of pseudo-labeling (Lee et al., 2013), *i.e.*, training models with unlabeled data and pseudo labels assigned by models' predictions.

As a widely used technique, the main problem of SSL is *how to generate accurate pseudo labels without or with tolerable effects of confirmation bias* (Arazo et al., 2020), *i.e.*, overfitting to incorrect pseudo labels from teacher models. There were three main strands of research, aiming at obtaining high-quality pseudo labels and a high sampling rate while being capable of various tasks and scenarios. Firstly, mainstream methods utilize threshold-based pseudo labeling (Sohn et al., 2020; Zhang et al., 2021; Kim et al., 2022; Wang et al., 2022b) with ad-hoc or complex hand-crafted strategies to *select high-quality pseudo labels*. However, these algorithms are predefined and task-specific, *i.e.*, they are designed for classification tasks but cannot handle more challenging regression tasks. The second strand introduces pre-trained teacher models (Zhou & Li, 2010; Xie et al., 2020b) to *generate high-quality pseudo labels*, which require extra computational cost (*e.g.*, double training times (Pham et al., 2021)) or suffer from confirmation bias (Yalniz et al., 2019). The third line explores consistency regulaizations (Xie et al., 2020a; Sohn et al., 2020; Li et al., 2021) to *prevent confirmation bias of inaccurate pseudo labels*, *e.g.,* optimizing the consistency loss with weak-strong augmentation, which only work for specific modalities with prior augmentations. Therefore, none of the previous SSL methods achieved three goals simultaneously.

This work answers a core question in SSL training: *how to efficiently evaluate a pseudo label comprehensively?* We introduce a reward score based on cosine similarity between pseudo and ground-truth labels as the quality standard, which is a smooth and well-calibrated metric for classification and regression tasks. Then, we propose a **Semi**-supervised **Reward** framework (**SemiReward**) that predicts reward scores based on pseudo labels and corresponding unlabeled data for pseudo-label selection and can be used as an add-on module for mainstream SSL methods. Specifically, a rewarder network predicts credible reward scores to filter pseudo labels for the student training and is learned to fit ground-truth reward scores online. To disentangle its training from the student, a two-stage training pipeline is designed with the assistance of a generator network, which generates "fake labels" that only train the rewarder. The rewarder and generator are first pre-trained alternatively on the labeled dataset in stage 1 to alleviate confirmation bias, then trained on a randomly subsampled set of labeled data and selected unlabeled data in stage 2. Empirical studies show that SemiReward predicts calibrated reward scores to select high-quality pseudo labels with a high sampling rate to boost SSL training. We conduct comparison experiments on SSL benchmarks with three modalities and two task types, verifying that SemiReward improves both general and modern SSL algorithms in performance and convergence speeds. Our main contributions are three folds:

- From a fresh perspective, we introduce the reward score to evaluate pseudo-label qualities and design the rewarder to predict it by modeling unlabeled data and pseudo-labels together.
- We propose a general and pluggable SemiReward framework that selects high-quality pseudo labels with reward scores. A two-stage training pipeline and a generator network are designed to train the rewarder online with negligible extra cost.

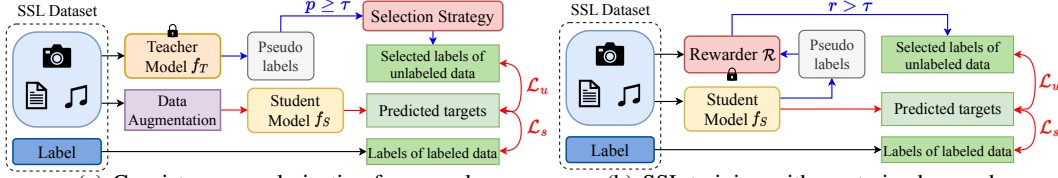

Figure 3: **Illustration of SSL training paradigm**, where blue lines denote pseudo-labeling pipeline and red lines denote gradient propagation. (a) Confidenced-based label selection strategy and strong-weak augmentations for consistency are task-specific and modality-specific (requiring ad-hoc augmentations). (b) Rewarder $\mathcal{R}$ is a plug-and-play label selection module for general SSL scenarios.

- Extensive experiments on 13 datasets validate that SemiReward markedly increases performance and convergence speeds of popular SSL methods in classification and regression tasks. We also empirically verify the reliability of reward scores and designed modules.

## 2 PRELIMINARY

**Semi-supervised training pipeline.** SSL is an extended scenario of SL, where given a labeled dataset $\mathcal{D}_L = \left\{x_i^l, y_i^l\right\}_{i=1}^{N_L}$ and an unlabeled dataset $\mathcal{D}_U = \{x_i^u\}_{i=1}^{N_U}$, with the sample numbers $N_L \ll N_U$. Considering any classification or regression task, $y_i^l \in \mathbb{R}^C$ denotes the encoded ground-truth label, where $C$ is the label dimension, and the model $f_S(\cdot)$ learns to predict $f_S(x) = y \in \mathbb{R}^C$. As for $C$-class classification, one-hot encoding is adopted for $y^l$ while converting the model output to $\arg\max \mathbf{p}(y)$. To utilize all training data, the general SSL training pipeline with pseudo-labeling contains three steps: **(a) Pseudo-label generation**. Given a teacher model $f_T(\cdot)$ that is well-trained on $\mathcal{D}_L$, it can generate pseudo-labels $y^u = f_T(x^u)$ for $\mathcal{D}_U$. **(b) Pseudo-label selection**. High-quality pseudo labels $\hat{\mathcal{D}}_U = \{\hat{y}^u\}^{\hat{N}_U} = \{\mathbb{I}(p_i^u, \tau)y_i^u\}_{i=1}^{N_U}$ are filtered by a label selection mechanism $\mathbb{I}(\cdot, \cdot)$, where $\tau \in [0, 1]$ is the threshold. **(c) Supervised and unsupervised losses computation**, denoted as $\mathcal{L} = \mathcal{L}_S + \mathcal{L}_U$. Given a mini-batch of $B_L$ data, $\mathcal{L}_S$ is written as:

$$\mathcal{L}_S = \frac{1}{B_L} \sum_{i=1}^{B_L} \mathcal{H}\Big(y_i^l, f_S\big(\omega(x_i)\big)\Big), \tag{1}$$

where $\omega(\cdot)$ denotes stochastic data augmentations and $\mathcal{H}(\cdot, \cdot)$ is the loss function used for the SL task, such as cross-entropy and $\ell_1$ loss for classification and regression tasks. Similarly, given a mini-batch of $B_U$ unlabeled data, taking popular consistency regularization frameworks (Sohn et al., 2020) as an example, the unsupervised loss is

$$\mathcal{L}_U = \frac{1}{B_U} \sum_{i=1}^{B_U} \mathbb{I}(p_i^u, \tau)\mathcal{H}\Big(\hat{y}_i^u, f_S\big(\Omega(x_i^u)\big)\Big), \tag{2}$$

where $\Omega(x_i^u)$ represents the strong augmented unlabeled data. As shown in Figure 3(a), the consistency regularization framework usually has three design aspects: (i) $f_T$ and $f_S$ share the same network architecture and parameters of $f_S$ are updated to $f_T$ by copying or exponential moving average (EMA). (ii) For most consistency-based SSL methods, a hand-crafted $\mathbb{I}(\cdot, \cdot)$ requires predicted classification confidence to distinguish reliable labels. (iii) Since the teacher $f_T$ is more reliable than the student $f_S$, the consistency that between $f_T$ and $f_S$ is introduced by constructing sample pairs $(\omega(x_i^u), \Omega(x_i^u))$ with strong-weak augmentations proposed by UDA (Xie et al., 2020a) and optimizing consistency through $\mathcal{L}_U$.

**Breaking Through Limitations of Confidence-based Label Selection.** Existing label selection strategies in step (ii) only use $y^u$ or the confidence $p^u$ to evaluate pseudo labels in hand-crafted policies, which cannot guarantee the quality and stability of $\hat{\mathcal{D}}_U$. Meanwhile, the designed steps (ii) and (iii) limit the task and modality generalities of the pseudo-labeling pipeline. To tackle these problems, we parameterize $\mathbb{I}(\cdot, \cdot)$ as a lightweight rewarder model $\mathcal{R}(x^u, y^u) = r$, where a reward score $r \in [0, 1]$ represents the label quality and is defined in Sec. 3.1. In Figure 3(b), the pre-trained $\mathcal{R}$ can evaluate the label quality comprehensively based on both $x^u$ and $y^u$, rather than solely depended on $y^u$. And we define $\mathcal{L}$ in a simple and general form:

$$\mathcal{L} = \underbrace{\frac{1}{B_L} \sum_{i=1}^{B_L} \mathcal{H}\Big(y_i^l, f_S\big(\omega(x_i)\big)\Big)}_{\mathcal{L}_L} + \underbrace{\frac{1}{B_U} \sum_{j=1}^{B_U} \mathbb{I}(\mathcal{R}(x_j^u, y_j^u) > \tau)\mathcal{H}\Big(\hat{y}_j^u, f_S\big(\omega(x_j^u)\big)\Big)}_{\mathcal{L}_U} + \mathcal{L}_{\mathrm{aux}}, \tag{3}$$

where $\mathcal{L}_{\mathrm{aux}}$ denote training losses of the rewarder $\mathcal{R}$ with generator $\mathcal{G}$ discussed in Sec. 3.2.

## 3   SEMIREWARD

Here, we introduce SemiReward for high-quality pseudo-label selection in general SSL tasks. In Sec. 3.1, we first define reward score as a pseudo-label evaluation metric and approximate it by a rewarder model. Then, Sec. 3.2 describes how to learn the rewarder through a two-stage pipeline.

### 3.1   MEASUREMENT OF LABEL QUALITY

Unlike popular ranking loss (Ouyang et al., 2022) in reinforcement learning (RL) (Schulman et al., 2017), we define a continuous metric of pseudo-label quality based on label similarity.

**Definition 3.1** (Reward Score). The reliability of a pseudo label $y^u$ of data $x$ is measured by label similarity $\mathcal{S}(\cdot, \cdot)$ with its ground truth label $y^l$, which can also be approximated by a rewarder $\mathcal{R}(\cdot, \cdot)$:

$$r(y^u, y^l) = \mathcal{S}(y^u, y^l) \simeq \mathcal{R}(x, y^u) \in [0, 1]. \tag{4}$$

The ideal reward score should satisfy *monotonicity* and *smoothness* (not increasing dramatically) and strive to meet the trend of *calibration* curve (Clark, 1975), where a lower reward confidence indicates poorer label quality. Therefore, we define the label similarity based on cosine similarity.

**Definition 3.2** (Label Similarity). Given vectorized label $y \in \mathbb{R}^C$, the label similarity between $y_i$ and $y_j$ is defined as scaled cosine similarity:

$$\mathcal{S}(y_i, y_j) = \frac{y_i \cdot y_j}{2 \, \|y_i\| \, \|y_j\|} + 0.5 \in [0, 1]. \tag{5}$$

Figure 4(a) verifies the properties of $r(y^u, y^l)$ by changing the label similarity metrics to negative $L_2$ distance and JS-divergence, and it shows that Eq. (5) can be the better choice.

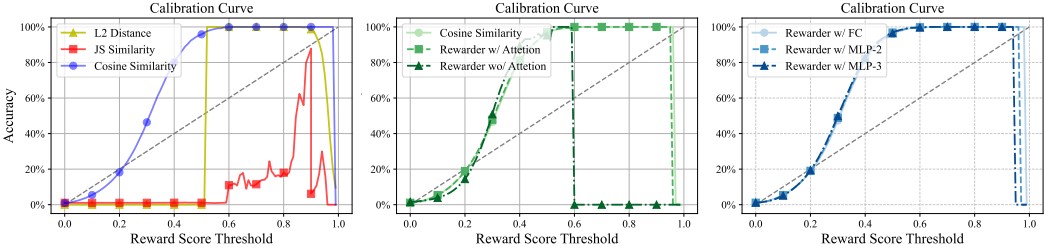

(a) Various reward similarities   (b) Attention module in Rewarder   (c) MLP module in Rewarder

Figure 4: **How rewarder works** illustrated by reward scores *v.s.* top-1 accuracy on CIFAR-100 (400 labels). (a) Analysis of alternative reward similarities; (b) Ablation of cross-attention module in $\mathcal{R}$, which is the vital component to learn calibrated reward scores; (c) Ablation of MLP layers.

To support both classification and regression tasks, we determine the encoding strategies to ensure that used labels are in vector format. This paper mainly discusses the cases of one-hot classification or single attribute regression. Given a raw scalar label, it can be encoded in "one-hot" format for classification. As for a raw regression label $y \in [0, C]$, we propose a soft one-hot encoding that equally divides the scalar into $C$ bins and sets the $k$-th position in the vector to $1 + (y - k)$, where $k \leq y < k+1$, while other positions are set to 0. Afterward, we verify Eq. (4) with regression tasks in Figure 5 and find that it can serve as a reliable metric and reduce the confirmation bias of raw pseudo labels. As for multi-label scenarios (Lin et al., 2017), we first encode raw labels for each task separately and then concatenate them as the final labels.

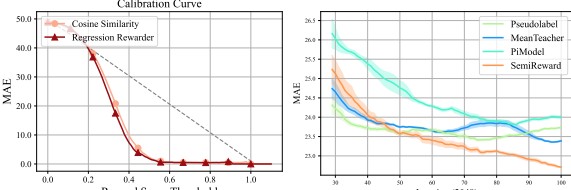

(a) Reward score *v.s.* MAE   (b) MAE *v.s.* training steps

Figure 5: Credible reward scores ensure the stable optimization of the student model, while raw pseudo labels in general SSL methods gravely misled the student for regression task on RCF-MNIST (1% labels).

**Rewarder**. As defined in Eq. (4), $\mathcal{R}(\cdot, \cdot)$ tries to solve a regression problem: the model should extract semantic information of $y^l$ from $x^u$ and tell the similarity between $x^u$ and $y^u$ according to their semantic correlation. As shown in Figure 6, $\mathcal{R}$ is designed as:

$$\mathcal{R}(x^u, y^u) = \text{Sigmoid}\Big(\text{MLP}\big(\text{CA}\big(\text{Emb}(f(x^u)), \text{Emb}(y^u)\big)\big)\Big), \tag{6}$$

where the input data and label are first linear embedded to the same dimension by $\mathrm{Emb}(\cdot)$, and their correlations are modeled by a cross-attention module $\mathrm{CA}(\cdot, \cdot)$ and a $\mathrm{MLP}(\cdot)$ module, then predict the reward score through Sigmoid function. Notice that $x^u$ is converted to last-layer features by a pre-trained backbone model $f(\cdot)$, $e.g.$, an image in $H \times W$ resolutions will be encoded as a $D$-dim feature $z^u \in \mathbb{R}^D$, which is easy for the network to capture high-level information directly related to $y^l$. As shown in Figure 4(b), we ablate modules in $\mathcal{R}$ and find that $\mathrm{CA}(\cdot, \cdot)$ is the most essential component to learn credible reward scores. Meanwhile, the backbone $f(\cdot)$ is also important to provide highly embedded features, or it will be hard and costly to learn such information by the lightweight $\mathcal{R}$. On the contrary, the number of layers in $\mathrm{MLP}(\cdot)$ has less impact on performance, as verified in Figure 4(c). As for implementation, $\mathcal{R}$ uses a 2-layer $\mathrm{MLP}(\cdot)$ with $D = 128$ and we simply apply the inherent teacher $f_T$ as $f(\cdot)$ in Eq. (6).

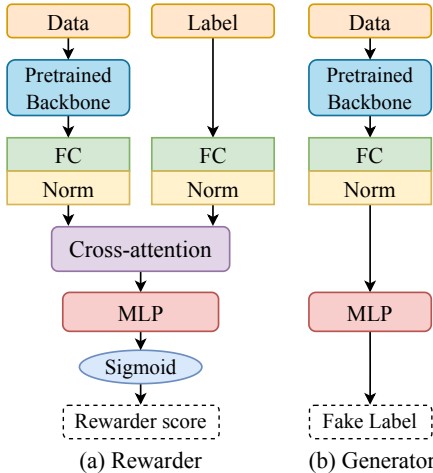

Figure 6: **Network structures** of efficient $\mathcal{R}$ and $\mathcal{G}$ analyzed in Table 5.

### 3.2 EFFICIENT TWO-STAGE TRAINING OF SEMIREWARD

Synchronizing with self-training paradigms, we train the rewarder $\mathcal{R}$ in a supervised manner with a reward training set $\mathcal{D}_R = \{\omega(x_i^r), y_i^r\}_{i=1}^{N_R}$, where $y^r$ is considered as the ground-truth label here. As discussed in Sec. 2, we expect a reliable $\mathcal{R}$ to filter pseudo labels to ensure high label quality to train $f_S$. Hence, we design a two-stage training paradigm for $\mathcal{R}$ in Figure 7, and $\mathcal{D}_R$ will be dynamically constructed by $\mathcal{D}_L$ and $\hat{\mathcal{D}}_U$. View Appendix B for a detailed analysis of training processes.

**Generator**. To train $\mathcal{R}$, we first design a generator $\mathcal{G}(x^u) = y^f \in \mathbb{R}^C$ to generate pseudo labels but not participate in the training process of $f_S$. Thus, we denote them as "fake labels". Similar to Chen et al. (2022a), $\mathcal{G}$ decouples the training of $f_S$ and $\mathcal{R}$ to avoid confirmation bias. Meanwhile, the fake labels generated by $\mathcal{G}$ gradually change from random to accurate, which helps $\mathcal{R}$ steadily fit reward scores on high-quality pseudo-label distributions. Its network is also as lightweight as $\mathcal{R}$, containing the pre-trained $f$ followed by a sample embedding $\mathrm{Emb}(\cdot)$ and a $\mathrm{MLP}(\cdot)$ module in Figure 6.

**Pre-training Rewarder**. $\mathcal{R}$ and $\mathcal{G}$ will be trained with fixed $\mathcal{D}_R = \mathcal{D}_L$ before $T$ training iterations. In the first stage, our main optimization goal is to approximate the ground truth reward scores with a wide range of fake labels without affecting the training of $f_S$. Thus, $\mathcal{R}$ does not select pseudo labels for the student $f_S$, and we introduce $\mathcal{G}(x^r) = y^f$ to generate fake labels that gradually get better. We compute losses for $\mathcal{R}$ and $\mathcal{G}$ alternatively as the auxiliary loss $\mathcal{L}_{\mathrm{aux}} = \mathcal{L}_\mathcal{R} + \mathcal{L}_\mathcal{G}$:

$$\mathcal{L}_\mathcal{R} = \frac{1}{B_R} \sum_{i=1}^{B_R} \ell_2 \Big( \mathcal{R}\big(x_i^r, \overline{\mathcal{G}}(x_i^r)\big), \mathcal{S}\big(y_i^r, \overline{\mathcal{G}}(x_i^r)\big) \Big), \tag{7}$$

$$\mathcal{L}_\mathcal{G} = \frac{1}{B_R} \sum_{i=1}^{B_R} \ell_2 \Big( \overline{\mathcal{R}}\big(x_i^r, \mathcal{G}(x_i^r)\big), 1 \Big), \tag{8}$$

where $\overline{\mathcal{R}}$ and $\overline{\mathcal{G}}$ denote forward without requiring gradients, which prevents two losses from interfering with each other. In implementations, we adopt two independent optimizers for $\mathcal{R}$ and $\mathcal{G}$ for

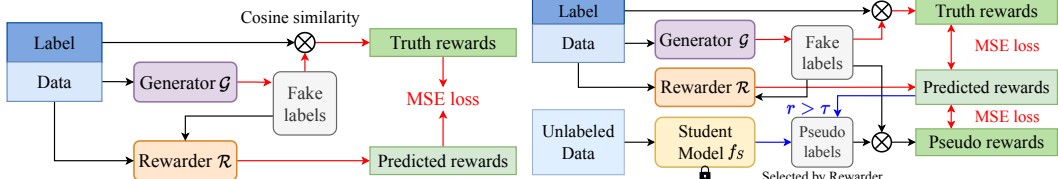

(a) Stage 1: Pre-training with labeled data      (b) Stage 2: Semi-supervised training with $\mathcal{D}_R$

Figure 7: **Two-stage training paradigm of SemiReward**. (a) To prevent $\mathcal{R}$ from distorting $f_S$, we pre-train $\mathcal{R}$ and $\mathcal{G}$ to convergence at the early stage of SSL training with $\mathcal{D}_L$. (b) After $T$ iterations, $\mathcal{R}$ further learns from the $\mathcal{D}_R$ sub-sampled from $\mathcal{D}_L \cup \hat{\mathcal{D}}_U$ with ignorable training cost.

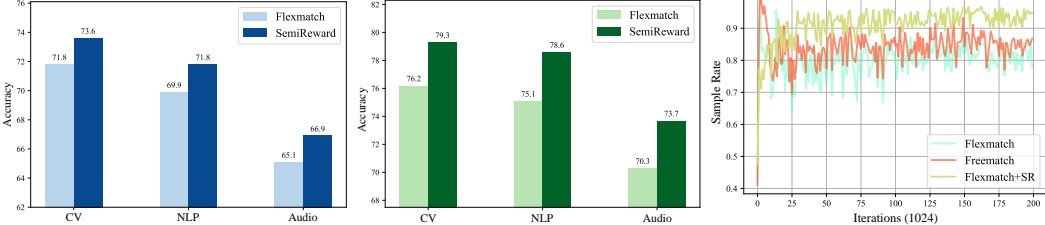

(a) Pseudo-label quality after stage 1     (b) Final pseudo-label quality     (c) Sampling rate *v.s.* training steps

Figure 8: **Evaluation of pseudo-label quality and quantity**. (a) After stage 1 in SemiReward training, it improves pseudo-label qualities by 1.7~2.1% over FlexMatch on CIFAR-100, Ag News, and UrbanSound-8k datasets. (b) SemiReward improves pseudo-label qualities of final models by 3.1~3.5% over FlexMatch. (c) Class-average sampling rate on STL-10 (40). Despite earning a high sampling rate at the very beginning of training in Flex/FreeMatch, it sustains below 90% during training. With SemiReward (SR), the sampling rate rapidly increases and remains 90~95%.

convenience, *e.g.,* Adam (Kingma & Ba, 2014). Therefore, $\mathcal{R}$ and $\mathcal{G}$ only run forward and backward once for rewarder training in each iteration, which costs ignorable extra overheads in SSL training.

**Semi-supervised training Rewarder**. In the second stage, the core objective is to optimize $f_S$ using $\mathcal{R}$ to filter high-quality labels as in Figure 3(b). As $f_S$ is continuously optimized on $\mathcal{D}_L \cup \hat{\mathcal{D}}_U$, $\mathcal{R}$ should also be efficiently optimized to suppress the confirmation bias in Pseudo Labeling. *i.e.,* $f_S$ is easily to overfit to incorrect pseudo-labels. We tackle this dilemma with a simple sub-sampling strategy: we further train $\mathcal{R}$ and $\mathcal{G}$ by Eq. (7) and Eq. (8) with randomly sub-sampled dataset $\mathcal{D}_R \subset \mathcal{D}_L \cup \hat{\mathcal{D}}_U$, where $N_R = \lambda(N_L + \hat{N}_U)$ and $\hat{\mathcal{D}}_U$ is the reliable pseudo-label set selected by $\mathcal{D}_R$. We adopt $\lambda = 0.1$ by default. This strategy combines two merits: (i) training $\mathcal{R}$ can be as fast as the first stage; (ii) similar to 10-fold cross-validation, exploring different subsets to train $\mathcal{R}$ avoids overfitting by introducing more randomness. As shown in Figure 8(c), SemiRewarder achieves high sampling rates compared to two confidence-based baselines, which select high-quality pseudo labels after stage 1 in Figure 8(a) and will maintain the high quantity in stage 2 as shown in Figure 8(b).

## 4 EXPERIMENTS

### 4.1 EXPERIMENTAL SETUP

**Comparison Methods for Classification.** In the context of classification tasks, we conducted experiments on 10 diverse datasets spanning three distinct modalities to assess the impact of integrating our SemiReward approach. All experiments are based on SSL benchmark USB (Wang et al., 2022a), which implement 14 SSL algorithms, including $\Pi$ model Rasmus et al. (2015), Pseudo Label Lee et al. (2013), Mean Teacher Tarvainen & Valpola (2017), VAT Miyato et al. (2018), MixMatch Berthelot et al. (2019b), ReMixMatch Berthelot et al. (2019a), UDA Xie et al. (2020a), FixMatch Sohn et al. (2020), Dash Xu et al. (2021), CoMatch Li et al. (2021), CRMatch Fan et al. (2021), FlexMatch Zhang et al. (2021), AdaMatch Berthelot et al. (2021), and SimMatch Zheng et al. (2022). We rigorously compare various SSL algorithms from them, Softmatch, Freematch, and Flexmatch, constituting the previous state-of-the-art, dubbed as **Previous SOTA**. Also, we choose the basic method Pseudo Label (Lee et al., 2013; Arazo et al., 2020) to illustrate the role of our approach in unlocking potential. Initially, we assess the performance of these algorithms based on classification error rates and training convergence speed, establishing a performance baseline. Subsequently, we can introduce SemiReward into the workflow and conduct a comparative analysis.

**Task Settings for Classification.** Here are tasks and specific settings on datasets of each modality. More information on datasets and experimental settings are detailed in Appendix A.1.

(a) For CV tasks, our investigations featured the deployment of renowned and challenging datasets, including CIFAR-100 (Krizhevsky et al., 2009), STL-10 (Coates et al., 2011), EuroSAT (Helber et al., 2019), and ImageNet (Deng et al., 2009), with the ImageNet pre-trained Vision Transformers (ViT) (Dosovitskiy et al., 2021) or randomly initialized ResNet-50 (He et al., 2016) architectures serving as the backbone.

(b) In the domain of NLP, we leveraged 3 datasets, including AG News (Zhang et al., 2015), Yahoo! Answers (Chang et al., 2008), and Yelp Review (yel, 2014), employing the self-supervised pre-trained Bert (Devlin et al., 2018) as the backbone.

Table 1: Top-1 error rate (%), performance gain, and training speedup times on nine SSL classification datasets with CV, NLP, and Audio modalities in various label settings.

| Domain | Dataset (Setting) | Pseudo Label | | FlexMatch | | SoftMatch/FreeMatch | | Average | |
|---|---|---|---|---|---|---|---|---|---|
| | | Base | +SR | Base | +SR | Base | +SR | Gain | Speed. |
| Audio | ESC-50 (250) | 38.42±0.85 | **33.33**±0.97 | 36.83±0.51 | **32.58**±0.51 | 32.71±0.82 | **29.71**±0.64 | **+4.11** | ×**1.73** |
| | ESC-50 (500) | 28.92±0.24 | **27.65**±0.32 | 27.75±0.41 | **25.92**±0.31 | 29.07±1.27 | **25.98**±0.49 | **+2.06** | ×**2.07** |
| | FSDnoisy18k (1773) | 34.60±0.55 | **33.24**±0.82 | 26.29±0.17 | **25.63**±0.28 | 29.39±1.83 | **26.10**±0.83 | **+1.77** | ×**1.30** |
| | UrbanSound8k (100) | 37.74±0.96 | **36.47**±0.65 | 37.88±0.46 | **36.06**±0.93 | 37.68±1.82 | **34.97**±1.02 | **+1.93** | ×**1.70** |
| | UrbanSound8k (400) | 27.45±0.96 | **25.27**±0.65 | 23.78±0.46 | **23.45**±0.93 | 23.78±0.13 | **19.39**±0.33 | **+2.30** | ×**1.08** |
| NLP | AG News (40) | 15.19±3.07 | **13.90**±0.21 | 13.08±3.94 | **12.60**±0.69 | 11.69±0.12 | **10.67**±0.90 | **+0.93** | ×**2.77** |
| | AG News (200) | 14.69±1.88 | **12.10**±0.58 | 12.08±0.73 | **11.05**±0.14 | 11.75±0.17 | **10.02**±0.82 | **+1.78** | ×**2.30** |
| | Yahoo! Answer (500) | 34.87±0.50 | **35.08**±0.40 | 34.73±0.09 | **33.64**±0.73 | 33.02±0.02 | **30.92**±0.90 | **+0.99** | ×**1.80** |
| | Yahoo! Answer (2000) | 33.14±0.70 | **32.50**±0.42 | 31.06±0.32 | **29.97**±0.10 | 30.34±0.18 | **29.11**±0.15 | **+0.99** | ×**3.53** |
| | Yelp Review (250) | 46.09±0.15 | **42.99**±0.14 | 46.09±0.15 | **42.76**±0.33 | 43.91±0.19 | **42.68**±0.12 | **+2.55** | ×**1.40** |
| | Yelp Review (1000) | 44.06±0.14 | **42.08**±0.15 | 40.38±0.33 | **37.58**±0.19 | 40.43±0.12 | **38.43**±0.14 | **+2.26** | ×**1.01** |
| CV | CIFAR-100 (200) | 32.78±0.20 | **31.94**±0.57 | 25.72±0.35 | **23.74**±1.39 | 21.07±0.72 | **20.06**±0.41 | **+1.28** | ×**1.04** |
| | CIFAR-100 (400) | 25.16±0.67 | **23.84**±0.20 | 17.80±0.57 | **17.59**±0.35 | 15.97±0.24 | **15.62**±0.71 | **+0.63** | ×**1.57** |
| | STL-10 (40) | 20.53±0.12 | **17.37**±0.47 | 11.82±0.51 | **10.20**±1.11 | 17.51±0.61 | **9.72**±0.62 | **+4.19** | ×**1.07** |
| | STL-10 (100) | 11.25±0.81 | **10.88**±1.48 | 7.13±0.20 | **7.59**±0.57 | 8.10±0.35 | **7.10**±1.39 | **+0.30** | ×**1.11** |
| | Euro-SAT (20) | 25.25±0.72 | **23.65**±0.41 | 5.54±0.16 | **4.86**±1.00 | 5.51±0.54 | **4.22**±0.34 | **+1.19** | ×**1.03** |
| | Euro-SAT (40) | 12.82±0.81 | **8.33**±0.33 | 4.51±0.24 | **3.88**±0.69 | 5.46±0.34 | **3.94**±0.71 | **+2.21** | ×**1.13** |

(c) For audio classification, we study the applications of SSL on 3 datasets, including Urban-Sound8k (Salamon et al., 2014), ESC-50 (Piczak, 2015), and FSDNoisy18k (Fonseca et al., 2019), where Hubert (Hsu et al., 2021) played the role of the pre-trained backbone.

**Comparison Methods and Task Settings for Regression.** To demonstrate the versatility of our approach, we extend our investigation to regression tasks alongside our primary focus. Specifically, we select Pseudo Label and its counterparts, namely the $\Pi$ model (Rasmus et al., 2015), CRMatch (Fan et al., 2021), and Mean Teacher (Tarvainen & Valpola, 2017), as our baseline methods. We then evaluate their performance in comparison to the integration of SemiReward on 3 regression datasets. The first two datasets, IMDB-WIKI (Rothe et al., 2018) and AgeDB (Moschoglou et al., 2017) with only 1% labeled data, perform face age regression. Additionally, we conduct a rotation angle estimation task using our custom RCF-MNIST dataset (Yao et al., 2022), featuring a more complex CIFAR-10 (Krizhevsky et al., 2009) background to align the samples closely to natural images and make the task more difficult. Experimental results are assessed based on two standard regression metrics: Mean Absolute Error (MAE) and Root Mean Square Error (RMSE).

**SemiReward Implementations.** To train the rewarder $\mathcal{R}$ and generator $\mathcal{G}$, we apply Adam (Kingma & Ba, 2014) optimizer with a fixed learning rate of 0.0005 in two-stage training for all tasks. We set the scheduler's $T$ to 10% of total SSL training iterations. During the inference process of $\mathcal{R}$, we use the *average reward score* as the threshold $\tau$ to filter pseudo labels dynamically. More specific hyperparameters are provided in Appendix A.2.

## 4.2 COMPARISON RESULTS ON SEMI-SUPERVISED BENCHMARKS

**Results on Classification.** Table 1 demonstrates the substantial performance improvements achieved by plugging SemiReward into representative SSL algorithms across diverse modalities, with notable impacts in audio-related tasks. When augmenting Pseudo Label with SemiReward, it outperforms SoftMatch on UrbanSound8k with 100 labeled instances and achieves an average performance gain of **4.11%** on ESC-50 with 250 labels. This enhancement effectively guides basic models, *e.g.*, Pseudo Label, toward more favorable local minima. The inclusion of SemiReward consistently expedites model convergence, as evidenced by the "avg. speedup" column in Table 1, with acceleration factors ranging from ×**1.5** to ×**3.53** in most cases. Total training times are shown in C.1. Meanwhile, the early stopping technique reduces training costs while maintaining desired performance, representing a valuable trade-off. Furthermore, using SemiReward can reduce training times and achieve lower error rates on Imagenet, as shown in Table 3. Notably, FlexMatch, in conjunction with SemiReward, surpasses previous SOTA methods, such as Freematch and Softmatch. The basic method with consistency regularization, FixMatch, also demonstrates substantial performance improvements when combined with SemiReward.

**Results on Regression.** We compare CRMatch, Mean Teacher, $\Pi$ model, Pseudo Label, and Pseudo Label added to SemiReward on RCF-MNIST, IMDB-WIKI, and AgeDB. The results are reported in Table 2. From the results of RMSE and MAE, SemiReward has great gain. Especially on RCF-

Table 2: RMSE and MAE, performance gain, and training speedup times on three SSL regression datasets with $1\%$ labels.

| Method | RCF-MNIST | | IMDB-WIKI | | AgeDB | |
|---|---|---|---|---|---|---|
| | RMSE | MAE | RMSE | MAE | RMSE | MAE |
| Supervised | $62.02_{\pm0.34}$ | $22.81_{\pm0.07}$ | $14.92_{\pm0.14}$ | $11.52_{\pm0.09}$ | $14.51_{\pm0.13}$ | $11.77_{\pm0.27}$ |
| Pseudo Label | $62.72_{\pm0.11}$ | $23.07_{\pm0.05}$ | $14.90_{\pm0.22}$ | $11.44_{\pm0.53}$ | $14.76_{\pm0.12}$ | $11.71_{\pm0.53}$ |
| $\Pi$-Model | $63.24_{\pm0.63}$ | $23.54_{\pm0.63}$ | $14.80_{\pm0.12}$ | $11.35_{\pm0.12}$ | $14.76_{\pm0.14}$ | $11.92_{\pm0.09}$ |
| MeanTeacher | $63.44_{\pm0.32}$ | $23.25_{\pm0.13}$ | $15.01_{\pm0.64}$ | $11.66_{\pm0.32}$ | $14.99_{\pm0.99}$ | $12.07_{\pm0.48}$ |
| CRMatch | $101.66_{\pm0.84}$ | $85.45_{\pm0.72}$ | $22.42_{\pm0.23}$ | $18.77_{\pm0.43}$ | $20.42_{\pm0.10}$ | $17.11_{\pm0.49}$ |
| **PseudoLabel+SR** | $\mathbf{61.71_{\pm0.34}}$ | $\mathbf{22.45_{\pm0.05}}$ | $\mathbf{14.80_{\pm0.53}}$ | $\mathbf{10.91_{\pm0.12}}$ | $\mathbf{14.01_{\pm0.12}}$ | $\mathbf{10.77_{\pm0.22}}$ |
| Gain | **-0.90** | **-0.99** | **-0.10** | **-0.53** | **-0.75** | **-0.94** |

Table 3: Top-1 error rate (%), performance gain, and training speedup times on ImageNet with 100 labels per class.

| Method | Top-1 | Gain | Speedup |
|---|---|---|---|
| FixMatch | 43.66 | +0.00 | ×1.00 |
| **FixMatch+SR** | **41.72** | **+1.94** | **×1.98** |
| FlexMatch | 41.85 | +0.00 | ×0.00 |
| FreeMatch | 40.57 | +1.28 | ×1.50 |
| SoftMatch | 40.52 | +1.33 | ×1.46 |
| **FlexMatch+SR** | **40.36** | **+1.49** | **×2.35** |

MNIST dataset, SemiReward can yield lower RMSE to **0.9** and MAE to **0.99**, which is even better than the supervised baseline. On the contrary, CRMatch performs poorly on various data sets, inferior to other SSL baselines, indicating the strong effect of confirmation bias.

## 4.3 ANALYSIS AND ABLATION

This section presents experimental analysis to demonstrate the functionality of SemiReward.

**Contribution of Each Component.** We do extensive ablation experiments and place them in Appendix B and obtain the following observations: (i) The number of MLP layers has little impact on the model's performance. The key lies in the design of the attention mechanism. (ii) Table 4 shows that replacing the used MSE ($\ell_2$) loss with BCE loss will make it difficult for the rewarder to converge and achieve poor scoring performance. Also, we find a scheduler that exceeds the reasonable setting range will cause the rewarder to be trapped in the wrong direction. The empirical starting time $T$ can be 10%. (iii) Comparing the training objectives of several models, we find that cosine similarity helps form the correspondence between pseudo labels and scores. (iv) Using the mean of reward scores to dynamically adjust the threshold $\tau$ performs much better than a fixed value in Figure A1.

Table 4: Ablation of rewarder training. We search the stage-2 start timing $T$ in the two-stage scheduler and losses for Eq. (7) and Eq. (8) on CIFAR-100 (400).

| Scheduler | Loss | | Error |
|---|---|---|---|
| T | MSE | BCE | (%) |
| 0% | ✓ | | 19.65 |
| 5% | ✓ | | 17.89 |
| 10% | ✓ | | **16.65** |
| 10% | | ✓ | 17.66 |
| 15% | ✓ | | 16.82 |

**Simplicity of SemiReward.** Table 5 shows SemiReward is very streamlined regarding parameters and FLOPs based on ViT-S-P4-32 on the CIRFA-100 dataset. Compared with the student model, our model accounts for a very low proportion of the training process, only requiring **1.28%** and **0.056%** extra parameters and FLOPs and computing two times forward and one times backward propagation in each iteration.

Table 5: Analysis of parameters and computational overhead (MFlops) of the student model, Rewarder, and Generator.

| Model | Params. (M) | FLOPs (M) |
|---|---|---|
| Student Model | 21.7 | 607.9 |
| Rewarder | 0.140 | 0.198 |
| Generator | 0.137 | 0.139 |
| Proportion | **1.28%** | **0.056%** |

**Regression Tasks with SemiReward.** Existing consistency regularization methods are unsuitable for regression tasks, with CRMatch being the only open-source alternative. However, CRMatch consistently yields subpar results, primarily due to confirmation bias (Arazo et al., 2020). Simultaneously, we note that in imbalanced regression datasets like IMDB-WIKI and AgeDB, SemiReward encounters challenges in enhancing the selection of superior pseudo-labels, hampering improved model convergence. Conversely, in tasks with balanced data distributions, such as rotation angle estimation, SemiReward demonstrates notably superior performance. This phenomenon may be attributed to the inherent difficulty in accurately labeling data points located at the distribution's extremes in imbalanced datasets, leading to partial performance degradation in such scenarios.

## 5 RELATED WORK

Pseudo Label (Lee et al., 2013) pioneered the generation of artificial labels for unlabeled data with models trained on labeled data, followed by consistency regularization (Samuli & Timo, 2017) aiming to ensure consistent predictions for different views of the same data, which are two foundational techniques in SSL. However, confirmation bias (Arazo et al., 2020; Chen et al., 2022a) caused by inaccurate pseudo labels limits SSL performances. Subsequent works mainly address this problem from three aspects: (i) selecting high-quality pseudo labels, (ii) generating high-quality pseudo labels, (iii) enhancing the tolerance of inaccurate labels. View Appendix D for detailed backgrounds.

**Improving Quality of Pseudo-labeling.**   Confidence-based thresholding techniques (Xie et al., 2020a; Xu et al., 2021) are designed to determine high-confidential pseudo labels. FixMatch (Sohn et al., 2020) relies on a fixed threshold but limits usage of more unlabeled data and leads to imbalanced pseudo-labels. FlexMatch (Zhang et al., 2021) employs class-specific thresholds to alleviate class imbalance by reducing thresholds for challenging classes. SoftMatch (Chen et al., 2022c) explores a trade-off between pseudo-label quantity and quality with a truncated Gaussian function to weigh sample confidence. FreeMatch (Wang et al., 2022b) introduces adaptive confidence thresholds based on the model's learning state. Moreover, contrastive learning is applied to thresholding methods, *e.g.,* adaptive contraction of the class space in ShrinkMatch (Yang et al., 2023) and the semantic similarity for mutual calibration in SimMatch (Zheng et al., 2022). However, these methods broadly enhance classification tasks but are inapplicable in regression tasks. CR-Match (Fan et al., 2021) presents FeatDistLoss, which also works for regression but does not yield satisfactory results.

**Improving Tolerance of Inaccurate Labels.**   Early SSL models exhibit heightened sensitivity to low-quality pseudo-labels, necessitating the enhancement of the model's error tolerance and label quality. The $\Pi$ model (Rasmus et al., 2015) introduces dual perturbations to input samples, while Temporal Ensembling (Samuli & Timo, 2017) maintains an EMA of label predictions for each training example. Mean Teacher (Tarvainen & Valpola, 2017) takes a step further by averaging model weights, reducing label dependency during training. Meanwhile, another line of research assumes the labeled datasets contain noisy labels and designs robust training strategies to discriminate inaccurate labels (Xu et al., 2021; Li et al., 2019a). Unlike them, SemiReward employs a two-stage training approach to learn reward scores, separating rewarder and student model training.

**Reward Modeling**   A reward function is crucial in conveying complex objectives to agents in reinforcement learning (RL) (Christiano et al., 2017). Most reward models (Leike et al., 2018) are supervised by classification losses, *e.g.*, ranking loss (Bradley & Terry, 1952), on constructed preference datasets from users. SURF (Park et al., 2022) adopts confidence-based pseudo-labeling to learn a reward function for preference-based RL. Recently, InstructGPT (Ouyang et al., 2022) provided a fine-tuning paradigm for aligning pre-trained large-scale language models (LLM) to human preference. However, reward modeling is designed and used for RL optimizations (Schulman et al., 2017) but has not been introduced to SSL scenarios.

## 6   CONCLUSION AND LIMITATION

**Contributions and Social Impacts**   This paper introduces SemiReward, a general and pluggable framework for SSL scenarios that evaluates and selects high-quality pseudo labels to boost the performance and convergence speeds of self-training techniques. The core idea is to select accurate pseudo labels by a reward score reflecting pseudo-label quality based on unlabeled data and pseudo labels. To achieve this, a simple but efficient rewarder network is designed to model correlations and predict credible reward scores, which is trained online in a two-stage pipeline assisted by a generator network to avoid confirmation bias. Extensive experiments on diverse classification and regression datasets demonstrate consistent performance gains and convergence speedup when applying SemiReward to popular SSL algorithms. We believe that SemiReward will be regarded as a new paradigm for measuring pseudo-label quality compared to previous confidence-based strategies and will inspire the SSL community to design effective methods in many application scenarios.

**Limitations and Future Works**   We hope this work might be valuable and inspire the SSL community and list some limitations and future directions: (1) The defined reward scores and rewarder only support sample-level labels, while fine-grained labels have been widely used in many scenarios requiring token-level rewarding, *e.g.*, object detection (Liu et al., 2021). (2) Despite the rewarder predicting a reliable indicator for high-quality labels, it requires repeating the teacher model and the rewarder several times to get reliable pseudo labels (discussed in Appendix B.3). It costs extra computational costs and might lead to performance decreasing at the end of training in Figure 2. We may further design a more efficient sampling and selection pipeline for SSL training. (3) In real-world scenarios, it might be useful to pre-train a general rewarder with large-scale pre-trained backbones on open-source datasets (Yalniz et al., 2019). Then, transfer it to specific SSL downstream tasks. (4) In RL scenarios, SemiReward might be useful to popular RLHF (Christiano et al., 2017; Ouyang et al., 2022) and LLM instruction alignment tasks, combining SSL with reward modeling for RL training as Park et al. (2022). (5) Extending SemiReward with adaptive data augmentations, *e.g.,* automatic mixup (Liu et al., 2022; Qin et al., 2024), to further enhance SSL performance.

ACKNOWLEDGEMENT

This work was supported by National Key R&D Program of China (No. 2022ZD0115100), National Natural Science Foundation of China Project (No. U21A20427), and Project (No. WU2022A009) from the Center of Synthetic Biology and Integrated Bioengineering of Westlake University. This work was done when Weiyang Jin, Zedong Wang, and Fang Wu interned at Westlake University. We thank the AI Station of Westlake University for the support of GPUs and all anonymous reviewers for polishing the writing of the manuscript.

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

APPENDIX

The appendix is structured as follows:

(A) In Appendix A, we provide implementation details are provided including dataset settings, hyperparameter settings, and training schedule.

(B) In Appendix B, we describe extensive ablation studies presented analyzing the impact of different architectural choices, training techniques, and loss functions.

(C) In Appendix C, we provide additional experimental results, including detailed training time statistics across different datasets and settings.

(D) In Appendix D, we further provide extensive related work to highlight connections and differences to the proposed approach.

(E) In Appendix E, we provide pseudocode for training pipelines of SemiReward.

## A  IMPLEMENTATION DETAILS

### A.1  DATASET SETTING

For a fair comparison, we train and evaluate all methods with the same ViT backbones and hyper-parameters in Table A3. As for CV, we evaluate SemiReward on common benchmarks: CIFAR-100 (Krizhevsky et al., 2009), Euro-SAT (Helber et al., 2019), STL-10 (Coates et al., 2011), and ImageNet (Deng et al., 2009) for image modality. Euro-SAT contains Sentinel-2 satellite images covering 13 spectral bands, which is not a natural image dataset as the other three. As for NLP, AG News (Zhang et al., 2015) (news topic material), Yahoo! Answer (Chang et al., 2008) (topic classification), and Yelp Review (yel, 2014) (sentiment classification) to evaluate SSL algorithms on more fine-grained sentiment NLP classification tasks. For audio classification, we choose Urban-Sound8k (Salamon et al., 2014) with a maximum length of 4 seconds, ESC-50 (Piczak, 2015) with a maximum length of 5 seconds, and FSDNoisy18k (Fonseca et al., 2019) with the length between 3 seconds and 30 seconds.

Table A1: Settings and details classification datasets in various modalities.

| Domain | Dataset | #Label per class | #Training data | #Validation data | #Test data | #Class |
|---|---|---|---|---|---|---|
| | CIFAR-100 | 2 / 4 | 50,000 | - | 10,000 | 100 |
| CV | STL-10 | 4 / 10 | 5,000 / 100,000 | - | 8,000 | 10 |
| | EuroSat | 2 / 4 | 16,200 | - | 5,400 | 10 |
| | ImageNet | 100 | 1,28,167 | - | 5,0000 | 1000 |
| | Yelp Review | 50 / 200 | 250,000 | 25,000 | 50,000 | 5 |
| NLP | AG News | 10 / 50 | 100,000 | 10,000 | 7,600 | 4 |
| | Yahoo! Answer | 50 / 200 | 500,000 | 50,000 | 60,000 | 10 |
| | ESC-50 | 5 / 10 | 1,200 | 400 | 400 | 50 |
| Audio | UrbanSound8k | 10 / 40 | 7,079 | 816 | 837 | 10 |
| | FSDnoisy18k | 52-171 | 1,772 / 15,813 | - | 947 | 20 |

Table A2: Settings and details of regression datasets in CV.

| Domain | Dataset | Task | #Label arrange | #Training data | #Validation data |
|---|---|---|---|---|---|
| | RCF-MNIST | Rotation | $[0, 360]$ | 50,000 | 10,000 |
| CV | IMDB-WIKI | Face age | $[1, 101]$ | 167,562 | 23,938 |
| | AgeDB | Face age | $[1, 101]$ | 106,750 | 15,250 |

We conducted age regression experiments on two datasets, IMDB-WIKI (Rothe et al., 2018) and AgeDB (Moschoglou et al., 2017) with 1% labels. AgeDB contains images of various celebrities, such as actors, writers, scientists, and politicians, and each image is annotated with identity, age, and gender attributes. The minimum and maximum ages are 1 and 101, respectively. The IMDB-WIKI dataset contains around 167,562 face images. Each image has an age and gender label associated with it, and the age range is 1∼101. The task here is to extract human features so that the model returns a continuous real value to predict age. Furthermore, we performed a rotation angle estimation task on our custom RCF-MNIST (Yao et al., 2022) dataset, which features a more intricate background CIFAR-10 (Krizhevsky et al., 2009), rather than the simple three-color backgrounds, to align the dataset's images more closely with natural images and make it more difficult. This dataset

can be solved with rotation features of objects except for the background image, allowing the model to regress a rotation angle of the foreground object.

## A.2 Hyperparameter and Training Settings

**Basic Settings.** As for classification tasks, regarding hyperparameter settings of SSL classification benchmarks constructed in USB (Wang et al., 2022a), we adopted the original settings with pre-trained Transformers as the backbone and made a few adjustments to adapt to SemiReward, as shown in Table A3. The total training iterations are set to $2^{20}$, and an early stop technique is used for calculating the convergence times. Meanwhile, we use the full experimental settings in Flex-Match (Zhang et al., 2021) for ImageNet, which uses 100 classes per class with ResNet-50 as the backbone. All methods are trained from scratch by SGD (Loshchilov & Hutter, 2016) optimizer with a momentum of 0.9, a basic learning rate of 0.03, and a cosine learning rate decay as USB. Note that Semi-AVES (Su & Maji, 2020) uses $224 \times 224$ input resolutions and ViT-S-P16-224 with the labeled and unlabeled batch size of 32, and other settings are the same as STL-10. We apply $\ell_1$ loss as the basic regression loss. As for regression tasks, we follow CV settings in USB to construct similar experiment settings for IMDB-WIKI (Rothe et al., 2018) ($224 \times 224$ resolutions as Semi-AVES in USB), AgeDB (Moschoglou et al., 2017) (as Semi-AVES), RCF-MNIST (Yao et al., 2022) ($32 \times 32$ resolutions as CIFAR-100). All experiments are implemented with PyTorch and run on NVIDIA A100 GPUs, using 4GPUs training by default.

Table A3: Hyper-parameters and training schemes of SSL classification tasks based on USB.

| Domain | CV | | | NLP | | | Audio | | |
|---|---|---|---|---|---|---|---|---|---|
| Dataset | CIFAR-100 | STL-10 | Euro-SAT | AG News | Yahoo! Answer | Yelp-5 | UrbanSound8k | FSDNoisy | ESC-50 |
| Image Size | 32 | 96 | 32 | \multicolumn{3}{c}{$-$} | \multicolumn{3}{c}{$-$} | | |
| Max Length | \multicolumn{3}{c}{$-$} | \multicolumn{3}{c}{512} | 4.0 | 5.0 | 5.0 | | |
| Sampling Rate | \multicolumn{3}{c}{$-$} | \multicolumn{3}{c}{$-$} | \multicolumn{3}{c}{16,000} | | | |
| Model | ViT-S-P4-32 | ViT-B-P16-96 | ViT-S-P4-32 | \multicolumn{3}{c}{BERT-Base} | \multicolumn{3}{c}{HuBERT-Base} | | |
| Weight Decay | \multicolumn{3}{c}{5e-4} | \multicolumn{3}{c}{1e-4} | \multicolumn{3}{c}{5e-4} | | | |
| Labeled Batch size | \multicolumn{3}{c}{16} | \multicolumn{3}{c}{4} | \multicolumn{3}{c}{8} | | | |
| Unlabeled Batch size | \multicolumn{3}{c}{16} | \multicolumn{3}{c}{4} | \multicolumn{3}{c}{8} | | | |
| Learning Rate | 5e-4 | 1e-4 | 5e-5 | 5e-5 | 1e-4 | 5e-5 | 5e-5 | 5e-4 | 1e-4 |
| Layer Decay Rate | 0.5 | 0.95 | 1.0 | 0.65 | 0.65 | 0.75 | 0.75 | 0.75 | 0.85 |
| Scheduler | \multicolumn{9}{c}{$\eta = \eta_0 \cos(\frac{7\pi k}{16K})$} | | | | | | | | |
| Model EMA Momentum | \multicolumn{9}{c}{0.999} | | | | | | | | |
| Eval EMA Momentum | \multicolumn{9}{c}{0.999} | | | | | | | | |
| Weak Augmentation | \multicolumn{3}{c}{Random Crop, Random Horizontal Flip} | \multicolumn{3}{c}{$-$} | \multicolumn{3}{c}{Random Sub-sample} | | | |
| Strong Augmentation | \multicolumn{3}{c}{RandAugment(Cubuk et al., 2018)} | \multicolumn{3}{c}{Back-Translation (Xie et al., 2020a)} | \multicolumn{3}{c}{Random Sub-sample, Gain, Pitch, Speed} | | | |

Table A4: Hyper-parameters and training schemes of SemiReward for various tasks and modalities.

| Hyperparameter | Classification | | | Regression | | |
|---|---|---|---|---|---|---|
| | CV | NLP | Audio | RCF-MNIST | IMDB-WIKI | AgeDB |
| Threshold $\tau$ | \multicolumn{3}{c}{Average} | Top-k | \multicolumn{2}{c}{Average} | |
| Optimizer | \multicolumn{6}{c}{Adam} | | | | | |
| Learning rate | \multicolumn{6}{c}{0.0005} | | | | | |
| Loss | \multicolumn{6}{c}{MSE} | | | | | |
| Embedding dim. | \multicolumn{6}{c}{128} | | | | | |
| MLP Layer-number | \multicolumn{6}{c}{2} | | | | | |
| Schedule $T$ | \multicolumn{6}{c}{10% of total iterations} | | | | | |
| Sun-sampling $\lambda$ | \multicolumn{6}{c}{0.1} | | | | | |

**SemiReward Settings.** In Table A4, we provide detailed hyper-parameters and settings for SemiReward training. The two-stage online training of the rewarder $\mathcal{R}$ and generator $\mathcal{G}$ is trained by Adam (Kingma & Ba, 2014) optimizer with a learning rate of 0.0005 for all tasks, independent of the student model's optimization. For each training step after $T$ iterations, $\mathcal{R}$ infers once and selects high-quality pseudo labels for the student with the *average reward score* as the threshold $\tau$, except for using top-k highest pseudo-labels for RCF-MNIST with $k = 16$. The generator $\mathcal{G}$ utilizes a 4-layer MLP (only containing FC layers and ReLU) with 256, 128, and 64 hidden dimensions.

## B Ablation Study Details

### B.1 Calibration Curve

To explore the properties of our proposed reward score in Sec. 3.1, we visualize the correlation between ground truth or learned reward scores and the quality of pseudo labels according to the

concept of calibration curves (Clark, 1975) in Figure 4 and Figure 5. Our goal in Eq. 4 is to learn a mapping from pseudo-labels to scores, which can be approximately linear or positively correlated and will have good discrimination and reliability in evaluating pseudo-label qualities. The data will not be classified as particular points within a small range, leading to excessive random error interference.

Therefore, we plan to analyze the reward score from four aspects: the threshold of reward score screening, the pseudo-label accuracy, and the confidence of the reward score. Based on the direct proportional relationship, we explore whether the model can achieve the required effect under different module designs and use this to illustrate through ablation experiments and theoretical analysis. Concretely, for each pseudo label that passes through the rewarder, we will return a corresponding score value and calculate the accuracy of the pseudo-labels after different thresholds by setting thresholds for different score values to draw a graph. In Figure 4(a), when calculating similarity, there is a sudden accuracy drop near a threshold value close to 1. This is caused by adding epsilon in numerical calculations to prevent division by zero errors in PyTorch implementation.

## B.2 Network Architecture of Rewarder

From the model design perspective, our rewarder network mainly incorporates a cross-attention mechanism to extract the information interaction between labels and data. On the other hand, it uses several layers of MLP to deepen feature processing further. Therefore, we conducted ablation experiments on CIRFA-100 with 400 labels to explore the impact of these mechanisms.

As presented in Table A5, we find that the incorporation of the cross-attention mechanism within the architectural module exerts a profound influence on both the pace of convergence and the ultimate efficacy of the model. Subsequent to the integration of a more profound MLP, the performance of the rewarder in the context of SemiReward exhibits no statistically significant enhancements. Instead, it is discernible that the augmentation has engendered a deceleration in the training process. Drawing upon

Table A5: MLP number stands for the FC layers in the rewarder.

| attention | MLP | Accuracy(%) | iteration |
|:---:|:---:|:---:|:---:|
| ✓ | 1 | 83.35 | 100352 iters |
| ✓ | 2 | 83.26 | 129024 iters |
| ✓ | 3 | 83.32 | 145408 iters |
|  | 1 | 81.99 | 194559 iters |
|  | 2 | 82.20 | 194559 iters |
|  | 3 | 82.25 | 204799 iters |

our meticulous calibration curve analysis in Figure 4 and Figure 5, it becomes readily apparent that the attention mechanism assumes a paramount role in evaluating the intrinsic performance of the rewarder model and the holistic training regimen. Its profound impact is manifest in the capability to orchestrate a seamless and continuous spectrum for score mapping of pseudo labels, as opposed to engendering numerous isolated points that could precipitate a distortion in the alignment between accuracies and reward scores.

## B.3 Scheduler for SemiRewrd

We conducted experiments on the CIFAR-100 with 400 labels and ESC-50 with 250 labels datasets. The model training effects under different start timings were counted. Start timing represents the time node from pre-training (stage-1) to semi-supervised training (stage-2) of the rewarder, indicating that SemiReward will utilize high-quality pseudo labels to ensure the further convergence of the student model and itself. This is what we define as SemiRewards scheduler.

Table A6: The starting time is the comparison of the round in which training starts to the total rounds. At the same time, we measured the convergence time and accuracy.

| Start Timing | CV | | Audio | |
|:---:|:---:|:---:|:---:|:---:|
|  | Accuracy(%) | iteration | Accuracy(%) | iteration |
| 0% | 80.35 | 159743 iters | 62.86 | 96255 iters |
| 5% | 82.11 | 169984 iters | 65.59 | 65535 iters |
| 10% | 83.35 | 100352 iters | 67.42 | 38911 iters |
| 15% | 83.18 | 174080 iters | 67.10 | 69631 iters |

It can be seen that when switching at 0%, the model achieves poor results on the two data sets. However, there is experience value in the range of 5%-15%, and the robustness to nodes is maintained. In fact, for the scheduler selection of the model, the intuitive understanding is that turning

out of range is more likely to produce poor results. This is because premature means that the pre-training phase has not been completed, causing problems with the score mapping during the initial screening and causing subsequent online training to learn worse score targets. Too late will make the model converge slowly and easily fall into local optimality, making it difficult to achieve favorable performance in the early stage.

For the screening phase, we employed a multi-forward approach to generate multiple pseudo-labels for a given dataset, facilitating iterative screening. The parameter **decay** denotes the frequency of forward passes. In the subsequent stages, we introduced an annealing strategy, dynamically adjusting **decay** throughout the training process. Specifically, we divided the total training steps by the current iteration, rounding up the result as the updated number of forward passes. To underscore that the performance enhancement of our algorithm extends beyond the impact of **decay** alone, we augmented the baseline algorithm with

Table A7: Analysis of selecting pseudo labels on CIFAR-100 (400 labels) with or without decay. Top-1 accuracy (%) and the training speedup times are reported.

| Method | FlexMatch |
|---|---|
| Baseline | 82.12 ($\times$1.0) |
| +Decay | 79.42 ($\times$1.4) |
| Semireward | 82.90 ($\times$**2.7**) |
| +Decay | **83.25**($\times$2.2) |

multiple forward passes and conducted comparative experiments A7. Our findings revealed that the algorithm achieves peak performance when **decay** and reward-based screening collaborate.

### B.4 LOSS FOR SEMIREWRD

In the ablation experiment, we not only compared the results of replacing MSE ($\ell_2$) loss with BCE loss. We also changed the algorithm of SemiReward total loss. Initially, two independent losses were used for gradient backpropagation, but we also considered the impact of weighting on the overall model training. We conducted ablation experiments on CIRFA-100 with 400 labels to compare their difference and find that the proposed MSE loss yields the best results.

As shown in Table A8, we can find that the weighted loss is more negative for model training, which may cause the rewarder to not converge and introduce many low-quality labels into the training process. Therefore, the importance of independent loss design can be seen here. On the other hand, BCE loss is also difficult to train the rewarder to convergence. This may be because our scoring model essentially follows the idea of regression tasks.

Table A8: Analysis of the loss types and loss weight for the proposed reward loss.

| MSE | BCE | Weighted | Accuracy(%) | iteration |
|---|---|---|---|---|
| $\checkmark$ | | $-$ | 83.35 | 100352 iters |
| $\checkmark$ | | 0.1 | 80.99 | 204799 iters |
| $\checkmark$ | | 0.5 | 81.25 | 204799 iters |
| $\checkmark$ | | 0.9 | 79.85 | 204799 iters |
| | $\checkmark$ | $-$ | 82.34 | 153600 iters |
| | $\checkmark$ | 0.1 | 80.02 | 196608 iters |
| | $\checkmark$ | 0.5 | 81.11 | 194559 iters |
| | $\checkmark$ | 0.9 | 81.01 | 196608 iters |

### B.5 TARGET FOR SEMIREWRD

As for the reward score, *i.e.*, the target of the rewarder model, its distance measurement is essential. We pursue that the scored pseudo-standards can be distributed evenly on the accuracy-score mapping with favorite properties mentioned in Sec. 3.1. Therefore, we constructed different score labels using different distance measures to train the rewarder and inferred why cosine similarity is an acceptable distance measure. We conducted ablation experiments on CIRFA-100 with 400 labels to compare the differences. As analyzed of Sec. 3, it can be seen that the divergence method represented by JS

Table A9: Analysis of the impact of training scoring targets calculated using different distance metric methods on the model, including using L2 distance and cosine similarity or not in SemiReward.

| Cosine Similarity | L2 Distance | Accuracy(%) | iteration |
|---|---|---|---|
| $\checkmark$ | | 83.35 | 100352 iters |
| | $\checkmark$ | 80.23 | 202751 iters |
| $-$ | $-$ | 82.25 | 204799 iters |

divergence has serious failures in the thinking of the calibration curve. This is because JS divergence may cause the scores of some tags to be too concentrated so that bad labels with similar scores will be selected as reliable labels. In Table A9, we found that the target score derived from the negative $L_2$ distance will cause the filtering ability of the rewarder to decline rapidly so that many low-

quality labels are selected, causing the training process of the student model trapped in relatively low accuracy.

### B.6 THRESHOLD FOR SEMIREWRD

In Appendix A1, we ablate the thresholding strategy for SemiReward, which compares the average thresholding with several fixed threshold $\tau$ settings, including 0.5, 0.7, and 0.9. The red dotted line denotes the result of the average strategy. In the context of reward score threshold-based filtering, it becomes evident that the fixation of this threshold engenders a multitude of challenges. During the training of SSL, employing a static threshold for pseudo-label selection poses prominent challenges (Zhang et al., 2021). During the early epochs of training, a model is still in its nascent state of underfitting and unstable. Setting a high threshold during these phases can inadvertently discard a substantial portion of potentially informative pseudo-labels. Such an action can curtail the model's ability to learn from these early indicators, potentially decelerating the overall convergence trajectory.

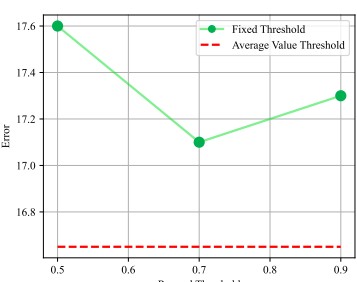

Conversely, as training progresses and the model refines its internal representations, a static low threshold may fall short in filtering out subpar-quality pseudo-labels. This introduces the hazard of the model overfitting these less reliable markers, jeopardizing its generalization capabilities. We advocate for a dynamic thresholding strategy grounded in averaging principles to address these challenges. Instead of adhering to a rigid threshold, our approach recalculates the threshold value within each mini-batch, considering the current quality distribution of the pseudo-labels. Such a mechanism ensures consistent retention of high-quality pseudo-labels throughout the training lifespan while effectively sidelining low-quality ones. Our empirical evaluations underline the efficacy of this method, not only amplifying the model's rate of convergence but also bolstering its performance on out-of-sample evaluations.

Figure A1: Thresholding $\tau$ for reward scores with adding SemiReward to FlexMatch on CIFAR-100 with 400 labels.

| Method | FlexMatch+SR |
|---|---|
| Coupled Training | 82.12 (×1.0) |
| +Gradient Ascent | 82.23 (×1.2) |
| Decoupled Training | 83.11 (×**2.2**) |
| +Gradient Ascent | **83.25**(×1.7) |

Table A10: Analysis of two training processes and the gradient accent of pseudo labels on CIFAR-100 (400 labels). Top-1 accuracy (%) and the training speedup times are reported.

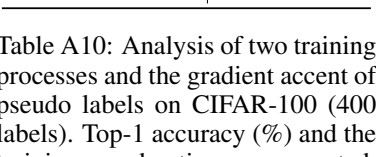

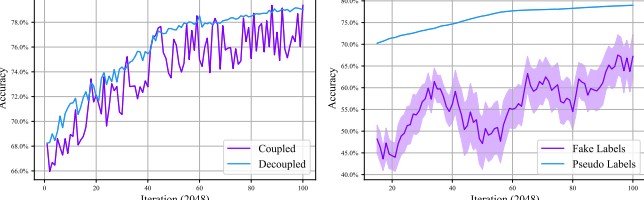

(a) Coupled *v.s.* Decoupled    (b) Pseudo labels *v.s.* Fake labels

Figure A2: Analysis of SR training on CIFAR-100 with Flex-Match. The mean and std of top-1 accuracy are plotted for (a) pseudo labels for the coupled and decoupled training and (b) pseudo and fake labels in the decoupled training.

### B.7 DECOUPLING OF STUDENT AND REWARDER TRAINING

As discussed in Sec. 3.2, we decouple the training of the student model and the rewarder by introducing the Generator and two-stage training pipeline to prevent confirmation bias. Here, we analyze the two training processes to verify whether the decoupled two-stage training with the Generator is an essential design. The first type of training process is to optimize the student and the Rewarder together without the Generator, where the teacher model generates candidate pseudo labels for the student and the Rewarder, which we call coupled training. Contrastively, the proposed two-stage training is the decoupled training. There are two reasons for decoupling the training process of the student and the Rewarder. Firstly, the Rewarder requires diverse pseudo-labels as the training data to fit the ground truth reward scores rather than deterministic high-performance labels. Secondly, the student and the Rewarder might suffer from confirmation bias. To further enhance the generated pseudo labels for the student training, we also designed a gradient ascent trick. Given selected reliable pseudo labels, we can modify them to generate more high-quality pseudo labels (or fake labels) by maximizing the reward scores with a step of gradient ascent in the inference process of the Rewarder.

Table A11: Training times and the average speedup times on nine SSL classification datasets with CV, NLP, and Audio modalities in various label settings.

| Modality | Dataset (Setting) | Pseudo Label | | FlexMatch | | SoftMatch/FreeMatch | | Avg. Speedup |
|---|---|---|---|---|---|---|---|---|
| | | Base | +SR | Base | +SR | Base | +SR | |
| Audio | ESC-50 (250) | 5.700 | 7.125×0.8 | 10.053 | 3.142×3.2 | 9.100 | 7.583×1.2 | ×1.73 |
| | ESC-50 (500) | 6.750 | 3.214×2.1 | 10.806 | 4.912×2.2 | 10.751 | 5.658×1.9 | ×2.07 |
| | FSDnoisy18k (1773) | 7.467 | 8.297×0.9 | 12.133 | 8.089×1.5 | 11.467 | 7.645×1.5 | ×1.34 |
| | UrbanSound8k (100) | 5.250 | 5.833×0.9 | 4.728 | 1.525×3.1 | 6.167 | 5.606×1.1 | ×1.70 |
| | UrbanSound8k (400) | 4.217 | 6.024×0.7 | 2.833 | 2.361×1.2 | 3.033 | 2.757×1.1 | ×1.08 |
| NLP | AG News (40) | 2.400 | 1.714×1.4 | 6.267 | 1.333×4.7 | 13.333 | 6.060×2.2 | ×2.77 |
| | AG News (200) | 2.889 | 1.699×1.7 | 3.556 | 1.693×2.1 | 4.444 | 1.434×3.1 | ×2.30 |
| | Yahoo! Answer (500) | 0.178 | 0.445×0.4 | 8.711 | 5.807×1.5 | 9.000 | 2.571×3.5 | ×1.80 |
| | Yahoo! Answer (2000) | 8.689 | 1.889×4.6 | 8.122 | 1.692×4.8 | 9.919 | 8.266×1.2 | ×3.53 |
| | Yelp Review (250) | 22.400 | 22.400×1.0 | 20.066 | 20.066×1.0 | 22.400 | 10.667×2.1 | ×1.39 |
| | Yelp Review (1000) | 1.822 | 4.673×0.4 | 21.411 | 16.470×1.3 | 19.133 | 16.394×1.2 | ×1.00 |
| CV | CIFAR-100 (200) | 9.320 | 11.314×0.8 | 54.280 | 49.345×1.1 | 54.889 | 49.899×1.1 | ×1.04 |
| | CIFAR-100 (400) | 14.920 | 13.564×1.1 | 100.240 | 45.564×2.2 | 94.044 | 67.174×1.4 | ×1.57 |
| | STL-10 (20) | 0.528 | 1.320×0.4 | 11.760 | 8.400×1.4 | 19.360 | 15.600×1.3 | ×1.07 |
| | STL-10 (40) | 0.268 | 0.693×0.4 | 9.556 | 7.351×1.3 | 20.267 | 13.889×1.5 | ×1.11 |
| | Euro-SAT (20) | 1.196 | 5.980×0.2 | 14.320 | 17.900×0.8 | 10.755 | 5.121×2.1 | ×1.03 |
| | Euro-SAT (40) | 1.092 | 5.460×0.2 | 21.040 | 23.378×0.9 | 16.800 | 7.304×2.3 | ×1.13 |

Table A12: Top-1 error rate (%), performance gain, and training speedup times on additional SSL classification datasets with CV and NLP modalities in various label settings.

| Domain | Dataset (Setting) | Pseudo Label | | FlexMatch | | SoftMatch/FreeMatch | | Average | |
|---|---|---|---|---|---|---|---|---|---|
| | | Base | +SR | Base | +SR | Base | +SR | Gain | Speed. |
| NLP | Amazon Review (250) | 53.45±1.90 | 49.13±0.77 | 45.73±0.11 | 43.08±0.11 | 45.29±0.95 | 42.98±0.24 | +3.09 | ×2.59 |
| | Amazon Review (1000) | 47.00±0.79 | 44.21±0.64 | 42.25±0.33 | 41.11±0.89 | 42.21±0.20 | 39.17±0.32 | +2.32 | ×2.92 |
| CV | Semi Aves 3959 (3959) | 40.35±0.3 | 37.93±0.45 | 32.48±0.15 | 31.23±0.09 | 32.85±0.31 | 31.02±0.15 | +1.82 | ×2.01 |
| | Tissuemnist (80) | 56.92±4.54 | 53.06±0.11 | 58.36±3.8 | 54.27±0.71 | 58.24±3.08 | 53.52±1.07 | +4.22 | ×1.92 |

As shown in Table B.6, when using the coupled training of the student and the Rewarder, Flex-Match+SR yields worse performance than the baseline (82.12 *vs.* 82.20), and FlexMatch+SR with the gradient ascent can only obtain a limited performance gain and speedup over the baseline. As shown in Figure 2(a), selected pseudo labels in the coupled training are unstable and affected by the student model, while the decoupled training produces high-quality pseudo labels steadily. Meanwhile, the proposed two-stage training decouples the student and the Rewarder by the Generator (aiming to maximize the reward score). It achieves a great trade-off between performance gains and speedup. Further applying the gradient ascent to the decoupled training will yield a little performance gain with more extra computational costs and cause unstable training. As shown in Figure 2(b), the quality of fake labels is relatively diverse, and it is difficult to obtain high-quality labels steadily. Therefore, we intend to use the decoupled training process without the gradient ascent trick as the final design.

# C   Extensive Experiment Results

## C.1   Details in Speedup

In Sec. 4, we give the average speed gain but not the specific training time. Table A11 gives the different training times corresponding to the nine sets of data sets in the three modes in the main text. We stipulate that the calculation is on a single NVIDIA A100 GPU to carry out relevant statistics, and the reported unit is the total hours.

## C.2   Capacity of SemiReward

From Table 1 and Table A11. In a few situations, SemiReward did not reach full convergence in a shorter time frame for primitive SSL algorithms like Pseudo Label, especially when evaluated on certain datasets such as STL-10 and Euro-SAT. This may be attributed to the simplicity of those basic methods like pseudo-labeling and entropy regularization in SSL tasks, which do not guide the model effectively towards a better local minimum. In contrast, our SemiReward compensates

for these shortcomings and unveils the potential of unlabeled data, allowing the model to progress toward better local minima, albeit requiring more time. This represents a trade-off and specific decisions about early stopping times for the optimal balance between speed and quality.

Table A13: Top-1 error rate (%), performance gain, and training speedup times on SSL classification datasets with CV in more label settings.

| Domain | CIFAR-100 (1000) | | CIFAR-100 (2500) | | CIFAR-100 (10000) | | Average | |
|---|---|---|---|---|---|---|---|---|
| | Flexmatch | +SR | Flexmatch | +SR | Flexmatch | +SR | Gain | Speed. |
| CV | $11.19_{\pm0.79}$ | $\mathbf{9.94_{\pm0.23}}$ | $10.82_{\pm1.90}$ | $\mathbf{9.42_{\pm0.66}}$ | $10.22_{\pm1.21}$ | $\mathbf{8.99_{\pm0.42}}$ | +1.29 | ×1.71 |

Table A14: Top-1 accuracy rate (%) and performance gain on ImageNet with 1% and 10% labels.

| Dataset | Label Settings | FixMatch | | CoMatch | | SimMatch | Average |
|---|---|---|---|---|---|---|---|
| | | Base | +SR | Base | +SR | Base | Gain |
| Imagenet | 1% | 53.5 | **55.1** | 66.0 | **67.4** | 67.2 | **+1.5** |
| | 10% | 71.6 | **72.8** | 73.6 | **74.5** | 74.4 | **+1.1** |

## C.3 RESULTS FOR ADDITIONAL DATASETS AND MORE LABEL SETTINGS

Due to the relatively antiquated nature and lower quality of the STL-10 dataset, our approach did not achieve optimal mean gain while emphasizing speed and lightweight characteristics. This can be attributed to the fact that we selected different random seeds multiple times, resulting in varied averages. Consequently, we have supplemented our study with datasets from the CV and NLP domains that exhibit superior performance in A12. In several settings of CIFAR-100, we have augmented the relevant tasks, as illustrated in A13, with the ImageNet pre-trained Vision Transformers (ViT) architecture serves as the backbone. Additionally, we have supplemented the data results for 1% and 10% labeled datasets (*i.e.*, 13 and 128 labels per class) in A14. We find that applying the proposed SemiReward (+SR) upon FixMatch (Sohn et al., 2020) and CoMatch (Li et al., 2021) can achieve around 1.3% performance gains, and CoMatch+SR outperforms the current SOTA SimMatch (Zheng et al., 2022)).

# D EXTENSIVE RELATED WORK

## D.1 SELF-TRAINING

In semi-supervised learning (SSL), self-training frameworks (Rosenberg et al., 2005; Grandvalet & Bengio, 2004; Yarowsky, 1995) play a very important role in unlabeled data utilization. Then, pseudo-labeling (Lee et al., 2013), as one of the classic self-training ways, pioneered the generation of artificial labels for unlabeled data. However, this embodiment faces the need for high-quality labels due to the problem of confirmation bias (Arazo et al., 2020). Subsequent work will mainly address this problem from two perspectives: one is to design a class or combine multiple methods to improve the quality of pseudo-label generation and application, and the other is to consider enhancing the network's acceptance of pseudo-labels, that is, a small number of low-quality pseudo-labels will not affect the overall prediction of the network.

**Consistency Regularization.** Samuli & Timo (2017) first proposed consistency regularization to ensure consistent predictions for similar data points, which has become a basic method for generating high-quality pseudo labels. Based on this, MixMatch (Berthelot et al., 2019b) and its variants (Berthelot et al., 2019a; Liu et al., 2023) performs data augmentation on unlabeled data, inputs multiple data into the same classifier, obtains different predicted classification probabilities, and uses a class method to make the average variance of multiple probability distributions smaller. UDA (Xie et al., 2020a) goes a step further and starts to use two branches of weak and strong augmented samples and regards the predictions of the weak augmentation branch as the target of the strong augmentation branch to improve the consistency of the pseudo-label and predictions. After that, ReMixMatch (Berthelot et al., 2019a) uses the distribution alignment method to encourage the marginal distribution of predictions for unlabeled data to be close to the marginal distribution of ground truth labels. Fixmatch (Sohn et al., 2020) designs a fixed confidence threshold to filter pseudo labels so that the high-quality pseudo-labels can be used in the SSL training process. The following works, like FlexMatch (Zhang et al., 2021), deeply explore the idea of confidence thresholds and

propose curriculum learning to dynamically adjust the thresholds generated by pseudo labels based on the training process. Additionally, softmatch (Chen et al., 2022c) shows the trade-off between the quantity and quality of pseudo labels and also derives a truncated Gaussian function to weight sample confidence. Freematch (Wang et al., 2022b) proposes a free matching method that adaptively adjusts confidence thresholds based on the model's learning state. The above methods essentially follow the strategy of training teacher-student distillation. Even the most advanced methods still rely on the manual design of confidence thresholds for screening. Although Meta Pseudo Labels (Pham et al., 2021) proposes to generate more accurate pseudo labels with a meta learner through bi-level optimization, it doubles training times and requires large-scale teacher models. This is why we proposed SemiReward as a simple but efficient solution for pseudo-label selection.

**Tolerance to Inaccurate Pseudo Labels.** Early SSL models have a certain sensitivity to low-quality pseudo labels. Then, another aspect of work starts by improving the model's tolerance to errors or low-quality labels. Π-Model (Rasmus et al., 2015) adds two different perturbations to an input sample, inputs the network twice to get the result, and then compares the consistency of the two results. This weakens the impact of low-quality labels but may be less efficient since two forward propagations are required to calculate the loss. Based on this, Temporal Ensembling (Samuli & Timo, 2017) maintains an EMA of label predictions on each training example and penalizes predictions that are inconsistent with this goal. Mean Teacher (Tarvainen & Valpola, 2017) further averages model weights instead of label predictions. This allows the use of fewer labels than sequential integration during training and also improves the accuracy of testing. Meanwhile, another branch of research assumes the labeled datasets are noisy and designs robust training or ad-hoc label selection policies to discriminate inaccurate labels (Xu et al., 2021; Li et al., 2019a; Tan et al., 2021).

## D.2 DISAGREEMENT-BASED MODELS

From the view of disagreement SSL, it is required to train two or three different networks simultaneously and label unlabeled samples with each other (Zhou & Li, 2010) so that they are less affected by model assumptions and loss functions. Co-training (Blum & Mitchell, 1998) assumes that each data point has two different and complementary views, and each view is sufficient to train a good classifier. Noisy Student (Xie et al., 2020b) is assigned pseudo-labels by a fixed teacher from the previous round, while (Yalniz et al., 2019) scales up this training paradigm to billion-scale unlabeled datasets. MMT (Ge et al., 2019), DivideMix (Li et al., 2019a) learn through multiple models or classifiers through online mutual teaching. Multi-head Tri-training (Ruder & Plank, 2018) uses training to learn three classifiers from three different training sets obtained using bootstrap sampling. In these methods, each classifier head is still trained using potentially incorrect pseudo-labels generated by other heads. Afterward, the classifier for pseudo-labels generated by DST (Chen et al., 2022b) is trained with unused pseudo-labels, thus having better tolerance to inaccurate pseudo-labels.

## D.3 SELF-SUPERVISED LEARNING FOR SSL

Self-supervised learning Xie et al. (2022); Li et al. (2023b; 2022; 2023a) techniques like contrastive learning (CL) approaches (Chen et al., 2020; He et al., 2020) are also widely applied to SSL, such as CoMatch (Li et al., 2021) that first introduced CL to the consistency regularization framework. ShrinkMatch (Yang et al., 2023) allows the model to search for contracted class space adaptively. In detail, for each uncertain sample, ShrinkMatch dynamically defines a shrunk class space, including the original top-1 class and less likely classes. Similarly, SimMatch (Zheng et al., 2022) uses semantic and instance similarity for mutual calibration. It uses the labeled data to train a semantic classifier and uses this classifier to generate pseudo labels for the unlabeled data. Meanwhile, ReMixMatch (Berthelot et al., 2019a) and CR-Match (Fan et al., 2021) utilize rotation prediction as the auxiliary task for SSL. Moreover, fine-tuning a pre-trained model on labeled datasets is a widely adopted form of transfer learning (TL), and several recent works (Li et al., 2018; 2019b; You et al., 2020; Ximei et al., 2021) like Self-Tuning (Ximei et al., 2021) combining TL with SSL methods. Self-Tuning (Ximei et al., 2021) and HCR (Tan et al., 2022) introduce CL pre-trained models as the regularization to mitigate confirmation bias in TL.

## D.4 ADVERSARIAL TRAINING FOR SSL

In the realm of SSL, innovative approaches have emerged that utilize adversarial training. One approach involves generating synthetic data (Odena, 2016; Dai et al., 2017) using a generator network and assigning it to a new "generated" class. The goal is to make the discriminator network provide class labels for these synthetic samples. Another line of research creates adversarial examples through techniques like VAT (Miyato et al., 2018), which adds noise to input data; VAdD (Park et al., 2018), introducing an adversarial exit layer into the model's architecture; and RAT (Suzuki & Sato, 2020), extending the concept of noise to input transformations. These methods aim to impose local smoothness constraints on the model's learned representations without relying on pseudo-labels during training. These advancements enhance model robustness and generalization, particularly in data-scarce scenarios, by utilizing latent data distribution structures for more effective learning. This research contributes significantly to improving SSL algorithms, addressing challenges in leveraging unlabeled data to enhance the applicability and performance of machine learning models in real-world applications. These innovative adversarial training approaches are poised to advance SSL.

# E    ALGORITHM

SemiRewards algorithm flow, including two-stage training (`SR_Train Stage 1` and `SR_Train Stage 2`) and inference (`SR_inference`), is as shown in Algorithm 1.

---

**Algorithm 1** Pseudocode of SemiReward training and inference in a PyTorch-like style.

---

```
# SR_Train Stage 1
iteration < T:
      # set SemiReward data loader
      for x_l,y_l in loader:
            x_r,y_r,B_R = x_l,y_l,B_l
  # load data in B_R size batch, x_r is labeled data and y_r is ground truth label
      for x_r,y_r,B_R in loader:
            feat(x_r) = f_s.feat(x_r) # get feature
            y_f = G(feat(x_r)) # get fake label

            r = S(feat(x_r),y_f) # get reward
            S = cossimin(y_r,y_f)) #get label similarity as targte
      # calculate loss
            L_R += MSE(r,S)
            L_G += MSE(r,1)
      L_aux = (L_R+L_G)/B_R
      # adam update
  L_aux.backward
      update(G)
      update(R)

# SR_Train Stage 2
iteration >= T:
      # set SemiReward data loader
      for x_u,x_l in loader:
            x_r = x_u+x_l

            # get pseudolabel y_p
            y_p = Pseudolabel(f_s(x_r))

            r = R(y_p,x_r)   # calculate reward for each pseudolabel in N

            # select top k reward in N
            sorted_indices = np.argsort(r)[::-1]
            y_p = y_p[sorted_indices]
            y_k = y_p[-k:]

            # get loader batch size B_R
            B_R = (B_l+B_u)*k/N

  # load data in B_R size batch, x_r is unlabeled data
      for x_r,y_k,B_R in sr_dataloader:
            y_f = G(x_r) # get fake label
            r = S(x_r,y_k) # get reward
            S = cossimin(y_k,y_f)) #get label similarity as targte
      # calculate loss
            L_R += MSE(r,S)
            L_G += MSE(r,1)
      L_aux = (L_R+L_G)/B_R
      # adam update
  L_aux.backward
      update(G)
      update(R)

# SR_Inference
iteration > T:
      for x_u,x_l,y_l in loader:
            # get pseudolabel y_p
            y_p = Pseudolabel(f_s(x_u))
            feat(x_u) = f_s+++.feat(x_u) # get feature

            r = R(feat(x_u),y_p) # evaluate score
            T = r.mean # get threshold
            mask_r = where(r>T,1,0)
            L_u = CrossEntropy(y_p,f_s(x_u))*mask # filter label
            L_l = CrossEntropy(y_l,f_s(x_l))
      # calculate loss
      L = L_u/B_U+L_l/B_L+L_aux # total loss
      # adamW update
  L.backward
      update(f_s)
```

---

`feat`: feature of input; `cossimin`: normalized cosine similarity; `cat`: concatenation.
`Pseudolabel`: pseudolabel method can see in Pseudo Label algorithm (https://arxiv.org/abs/1908.02983)

