# OpenReview forum: "SemiReward: A General Reward Model for Semi-supervised Learning"
_ICLR.cc/2024/Conference — ICLR 2024 poster_

### Official Review · Reviewer_DH5L · 2023-10-30

**Soundness:** 3 good
**Presentation:** 3 good
**Contribution:** 3 good
**Rating:** 6
**Confidence:** 5

**Summary:**

This paper studies the problem of semi-supervised learning, where the labeled data is limited and unlabeled data are massive during training. Previous pseudo labeling based methods are limited to pre-defined schemes or complex hand-crafted policies specially designed for classification. To address this, the authors present the Semi-supervised Reward framework (SemiReward), which assesses and chooses high-quality pseudo labels through predicted reward scores and is adaptable to mainstream SSL methods for both classification and regression tasks.

**Strengths:**

1. The paper is well-written and easy to understand.
2. The proposed integration of a reward model with an SSL model enhances pseudo label quality evaluation, with extensive experiments demonstrating its effectiveness across diverse tasks.
3. The experimental protocol is detailed and conducive to reproduce the results.

**Weaknesses:**

1. The method involves multiple models and training processes, adding complexity.
2. The manuscript would benefit from pseudo codes for improved clarity.
2. The design of reward model and hyper-parameter selection influence the outcomes, affecting the method's feasibility in practical scenarios.

**Questions:**

See weaknesses.

---

> ### Author Response · Authors · 2023-11-16
> **Response to Reviewer DH5L PART 1**
>
> **W1&W3: The method's complexity, involving multiple models and training processes and the design of the reward model and hyper-parameter selection, raises concerns about the method's practical feasibility.**
>
> **R:** From the complexity perspective, our algorithmic flow might be complex at first glance, but it is actually quite clear, concise, and user-friendly. We discuss the complexity of SemiReward from five aspects (for **W1**). *As for the complexity of user usage*, the most essential component in the inference process is the Rewarder, as described in the pseudocode (provided in PART 3) and Figure 3. It is a pluggable module to assess the quality of pseudo labels, subsequently filtering them based on predicted reward scores, which can be treated as previous pseudo-label selection strategies and utilized in various pseudo-label methods. *As for the two-stage training process*, it can be regarded as another line of self-teaching process involving two models (the Rewarder and the Generator). The role of the Generator is similar to the teacher model in the self-training pipeline to auxiliarily optimize the Rewarder. And, these two stages are actually the same process but use different input datasets, i.e., the second stage further utilizes the selected reliable pseudo labels $\hat{\mathcal{D}}_{U}$ in additional to the first stage. *As for the complexity of the model itself*, our Rewarder is relatively lightweight, and its architecture is easy to explain (as discussed in Sec. 3.1). The relevant parameters and computational costs of our model are provided in Table 5 of the main text as follows:
>
> |     Model     |          Params. (M)          |            FLOPs (M)           |
> |:-------------:|:-----------------------------:|:------------------------------:|
> | Student Model |              21.7             |              607.9             |
> |    Rewarder   |             0.140             |              0.198             |
> |   Generator   |             0.137             |              0.139             |
> |   Proportion  |**1.28** |**0.056** |
>
> (view PART 2 to continue)

---

> ### Author Response · Authors · 2023-11-16
> **Response to Reviewer DH5L PART 2**
>
> Furthermore, *as for the practical usage of SemiReward*, the results in Table A9 of the appendix indicate that the inclusion of our model significantly enhances the overall training speed of the student model. Note that the numbers in parentheses represent the acceleration factor. This indirectly underscores the lightweight and fast-convergence nature of our model (for **W3**).
>
> Table: Training speedup times on nine SSL classification datasets with CV, NLP, and Audio modalities in various label settings.
>
> | Modality | Dataset (Setting) | Pseudo Label Base | Pseudo Label +SR | FlexMatch Base | FlexMatch +SR | Soft/FreeMatch Base | Soft/FreeMatch +SR | **Avg. Speedup** |
> | ---------| -------------------| :----------------: | :---------------: | :------------: | :------------: | :-----------------------: | :---------------------: | :---------------: |
> | Audio    | ESC-50 (250)        | 5.700              | 7.125 **(×0.8)**  | 10.053         | 3.142 **(×3.2)**  | 9.100                    | 7.583 **(×1.2)**       | **(×1.73)**       |
> |          | ESC-50 (500)        | 6.750              | 3.214 **(×2.1)**  | 10.806         | 4.912 **(×2.2)**  | 10.751                   | 5.658 **(×1.9)**       | **(×2.07)**       |
> |          | FSDnoisy18k (1773)  | 7.467              | 8.297 **(×0.9)**  | 12.133         | 8.089 **(×1.5)**  | 11.467                   | 7.645 **(×1.5)**       | **(×1.34)**       |
> |          | UrbanSound8k (100)  | 5.250              | 5.833 **(×0.9)**  | 4.728          | 1.525 **(×3.1)**  | 6.167                    | 5.606 **(×1.1)**       | **(×1.70)**       |
> |          | UrbanSound8k (400)  | 4.217              | 6.024 **(×0.7)**  | 2.833          | 2.361 **(×1.2)**  | 3.033                    | 2.757 **(×1.1)**       | **(×1.08)**       |
> | NLP      | AG News (40)        | 2.400              | 1.714 **(×1.4)**  | 6.267          | 1.333 **(×4.7)**  | 13.333                   | 6.060 **(×2.2)**       | **(×2.77)**       |
> |          | AG News (200)       | 2.889              | 1.699 **(×1.7)**  | 3.556          | 1.693 **(×2.1)**  | 4.444                    | 1.434 **(×3.1)**       | **(×2.30)**       |
> |          | Yahoo! Answer (500) | 0.178              | 0.445 **(×0.4)**  | 8.711          | 5.807 **(×1.5)**  | 9.000                    | 2.571 **(×3.5)**       | **(×1.80)**       |
> |          | Yahoo! Answer (2000)| 8.689              | 1.889 **(×4.6)**  | 8.122          | 1.692 **(×4.8)**  | 9.919                    | 8.266 **(×1.2)**       | **(×3.53)**       |
> |          | Yelp Review (250)   | 22.400             | 22.400 **(×1.0)** | 20.066         | 20.066 **(×1.0)** | 22.400                   | 10.667 **(×2.1)**      | **(×1.39)**       |
> |          | Yelp Review (1000)  | 1.822              | 4.673 **(×0.4)**  | 21.411         | 16.470 **(×1.3)** | 19.133                   | 16.394 **(×1.2)**      | **(×1.00)**       |
> | CV       | CIFAR-100 (200)     | 9.320              | 11.314 **(×0.8)** | 54.280         | 49.345 **(×1.1)** | 54.889                   | 49.899 **(×1.1)**      | **(×1.04)**       |
> |          | CIFAR-100 (400)     | 14.920             | 13.564 **(×1.1)** | 100.240        | 45.564 **(×2.2)** | 94.044                   | 67.174 **(×1.4)**      | **(×1.57)**       |
> |          | STL-10 (20)          | 0.528              | 1.320 **(×0.4)**  | 11.760         | 8.400 **(×1.4)** | 19.360                   | 15.600 **(×1.3)**      | **(×1.07)**       |
> |          | STL-10 (40)          | 0.268              | 0.693 **(×0.4)**  | 9.556          | 7.351 **(×1.3)** | 20.267                   | 13.889 **(×1.5)**      | **(×1.11)**       |
> |          | Euro-SAT (20)        | 1.196              | 5.980 **(×0.2)**  | 14.320         | 17.900 **(×0.8)** | 10.755                   | 5.121 **(×2.1)**       | **(×1.03)**       |
> |          | Euro-SAT (40)        | 1.092              | 5.460 **(×0.2)**  | 21.040         | 23.378 **(×0.9)** | 16.800                   | 7.304 **(×2.3)**       | **(×1.13)**       |
>
> Lastly, *from the perspective of tunable hyperparameters*, our model actually incorporates fewer hyperparameters compared to current state-of-the-art methods such as FreeMatch and SoftMatch. We have fixed the network architectures of the Rewarders and Generator and only tuned three hyperparameters for the Rewarder training. Meanwhile, ablation and analysis experiments (e.g., Sec. 4.3 and Appendix B) have demonstrated the hyper-parameter robustness of our SemiReward, whereas manually designed methods often require extensive tuning for different task types. The comparison table of tunable hyperparameters is provided for reference (for **W3**):
>
> |  FlexMatch+SR  |  FreeMatch  |  SoftMatch  |
> | :-----: | :-----: | :-----: |
> | start_timing | label_hist | ema_p |
> | sr_lr | self-adaptive fairness (SAF) L_f | n_sigma |
> | N_k | w_u | per_class |
> | - | w_f | dist_align |
> | - | - | dist_uniform |
>
> (view PART 3 to continue)

---

> ### Author Response · Authors · 2023-11-16
> **Response to Reviewer DH5L PART 3**
>
> For instance, as demonstrated in the experiments outlined in Table A6 of the appendix, our model exhibits robustness within a certain range, as indicated by start_timing. Therefore, the user can apply the provided empirical hyperparameters to use SemiReward without hesitation (for **W3**).
>
> |      Start Timing     |      CV      |              |     Audio    |             |
> |:---------------------:|:------------:|:------------:|:------------:|:-----------:|
> |                       | Accuracy(\%) |   iteration  | Accuracy(\%) |  iteration  |
> |          0\%          |     80.35    | 159743 iters |     62.86    | 96255 iters |
> |          5\%          |     82.11    | 169984 iters |     65.59    | 65535 iters |
> | **10\%** |   **83.35**  | **100352** iters |   **67.42**   | **38911** iters |
> |          15\%         |     83.18    | 174080 iters |     67.10    | 69631 iters |
>
> **W2: Offer clarity-enhancing pseudo codes.**
>
> **R:** Thanks for your suggestions on improving the readability of our manuscript, and we provide the relevant pseudocode below. It has been incorporated into the appendix of the revision:
>
> ```
> # SR_Train Stage 1
> iteration < T:
> 	# set SemiReward data loader
> 	for x_l,y_l in loader:
> 		x_r,y_r,B_R = x_l,y_l,B_l
>   # load data in B_R size batch, x_r is labeled data and y_r is ground truth label
> 	for x_r,y_r,B_R in loader:
> 		feat(x_r) = f_s.feat(x_r) # get feature
> 		y_f,r = G(feat(x_r)),S(feat(x_r),y_r) # get fake label and reward
> 		S = cosine_similarity_n(y_r,y_f)) #get label similarity as targte
> 	# calculate loss
> 		L_R += MSE(r,S)
> 		L_G += MSE(r,1)
> 	L_aux = (L_R+L_G)/B_R
> 	# adam update
>   L_aux.backward
> 	update(G,R)
> # SR_Train Stage 2
> iteration >= T:
> 	# set SemiReward data loader
> 	for x_u,x_l in loader:
> 		x_r = cat(x_u,x_l)
> 		# get pseudolabel y_p, Pseudolabel method can see in Pseudo Label algorithm (https://arxiv.org/abs/1908.02983)
> 		y_p = Pseudolabel(f_s(x_r))
> 		r = R(y_p,x_r) # calculate reward for each pseudolabel in N
> 		y_k = Topk(y_p,r) # select top k reward in N by reward
> 		B_R = (B_l+B_u)*k/N # get loader batch size B_R
>   # load data in B_R size batch, x_r is unlabeled data
> 	for x_r,y_k,B_R in sr_dataloader:
> 		feat(x_r) = f_s.feat(x_r) # get feature
> 		y_f,r = G(feat(x_r)),S(feat(x_r),y_k) # get fake label and reward
> 		S = cosine_similarity_n(y_k,y_f)) #get label similarity as target
> 	# calculate loss
> 		L_R,L_G = MSE(r,S),MSE(r,1)
> 	L_aux = (L_R+L_G)/B_R
> 	# adam update
>   L_aux.backward
> 	update(G,R)
> # SR_Inference
> iteration > T:
> 	for x_u,x_l,y_l in loader:
> 		# get pseudolabel y_p, Pseudolabel method can see in Pseudo Label algorithm (https://arxiv.org/abs/1908.02983)
> 		y_p = Pseudolabel(f_s(x_u))
> 		feat(x_u) = f_s.feat(x_u) # get feature
> 		r = R(feat(x_u),y_p) # evaluate score
> 		mask_r = where(r>r.mean,1,0) # get threshold by T = r.mean
> 		L_u = CrossEntropy(y_p,f_s(x_u))*mask # filter label
> 		L_l = CrossEntropy(y_l,f_s(x_l))
> 	# calculate loss
> 	L =  L_u/B_U+L_l/B_L+L_aux # total loss
> 	# adamW update
>   L.backward
> 	update(f_s)
> ```

---

> ### Author Response · Authors · 2023-11-19
>
> Dear reviewer,
>
> We appreciate your valuable feedback. In our response, we added pseudocode-related content (more specific has been updated in the manuscript). At the same time, we have explained through discussions from three angles that our Rewarder is actually not complicated and does not reflect additional hyperparameters, and is very easy to follow due to its end-to-end and plug-and-play features.
>
> As you may know, unlike previous years, this year's discussion period only lasts until November 22nd, and we are getting closer to that deadline. We hope to discuss this with you in the meantime and will be happy to provide more information based on your feedback or further questions.
>
> If you are satisfied with our response, please consider updating your score. If you need any clarification, please feel free to contact us.

---

> > ### Comment · Reviewer_DH5L · 2023-11-20
> >
> > Thank you for your response. I have carefully reviewed the rebuttal provided, as well as the insights from other reviews. I‘ve decided to uphold my original score and raise my confidence from 4 to 5.

---

> ### Author Response · Authors · 2023-11-20
>
> Thank you for your reply and efforts in going through our response! Since the rebuttal period is approaching the end, feel free to discuss them with us if you have any other questions. We are glad to further improve the quality of our manuscript. Thanks again for your help!
>
> Best regards,
>
> Authors

---

### Official Review · Reviewer_uket · 2023-10-31

**Soundness:** 3 good
**Presentation:** 3 good
**Contribution:** 3 good
**Rating:** 6
**Confidence:** 5

**Summary:**

This paper proposes Semi-supervised Reward (SemiReward), an add-on module that uses a reward network and a generator network in a two-stage training pipeline, to train a student model with labeled and unlabeled data. The rewarder network predicts scores to filter pseudo labels for the student training, and the generator network generates fake labels (to provide the rewarder with a variety of scenarios, including incorrect ones, so that it learns to distinguish good data from bad more effectively without affecting the training of the student network) that only train the rewarder. The paper proposes a two-stage training pipeline: 1) pre-training both the rewarder network and a generator network on a labeled dataset, 2)  the reward network is trained on a smaller, random subset of the labeled data along with selected unlabeled data. Empirical results on NLP, Vision and Audio datasets show the effectiveness of their framework.

**Strengths:**

+ This paper is well-written and easy to follow, the motivation and formulation of the paper are sound

+ Figures and examples are informative and help understand the paper

+ Numerous ablation studies and qualitative comparisons help in understanding the setting

+ This paper shows strong empirical results that surpass current state-of-the-art methods on SSL

+ This paper adds non-trivial contributions and builds on prior work to present a novel framework for SSL

**Weaknesses:**

- Missing experiments: recent works show the effectiveness of their proposed methods by varying the number of labeled samples, given the sensitivity of available data and their impact, it seems imperative to include such experiments to asses the proposed method.

- Missing benchmarks: it is hard to asses the impact of the proposed framework without at least a large-scale experiment (e.g., ImageNet)

- Some gains, especially in NLP and CV datasets seem marginal (e.g., TL +0.30)

**Questions:**

- The NLP experiments are performed using 1\% labels, what is the labeled \% used in the CV and Audio datasets?

---

> ### Author Response · Authors · 2023-11-16
> **Response to Reviewer uket**
>
> **W1: Lack of experiments with samples of different labeling levels (missing experiments).**
>
> **R:** In several settings of CIFAR-100, we have augmented the relevant tasks. As illustrated in the table, the backbone for the student model is set as pretrained VIT:
>
> |  | CIFAR-100(1000) | CIFAR-100(2500) | CIFAR-100(10000) |
> | --- | --- | --- | --- |
> | FlexMatch | 11.19±0.79 | 10.82±1.90 | 10.22±1.21 |
> | FlexMatch+**SR** | **9.94±0.23** | **9.42±0.66** | **8.99±0.42** |
> | Avg. Gain| |  **1.29**   ||
>
> **W2: Lack of large-scale benchmarks like ImageNet (missing benchmarks).**
>
> **R:** Following FlexMatch and SoftMatch, we have provided results for ImageNet with 100 labels per class **in Table 3 of the main text**. Additionally, we have supplemented the data results for 1% and 10% labeled datasets (i.e., 13 and 128 labels per class) following CoMatch, as presented below. We find that applying the proposed SemiReward (+SR) upon FixMatch and CoMatch can achieve around 1.3% performance gains, and CoMatch+SR outperforms the existing SOTA method, SimMatch (CVPR’2022). Note that all these experiments train the student model from scratch.
>
> | Labels | FixMatch | FixMatch+**SR** | CoMatch | CoMatch+**SR** | SimMatch | Gain |
> | --- | :---: | :---: | :---: | :---: | :---: | :---: |
> | 1% | 53.5 | **55.1** | 66.0 | **67.4** | 67.2 | **1.5** |
> | 10% | 71.6 | **72.8** | 73.6 | **74.5** | 74.4 | **1.1** |
>
> **W3: Marginal gains in NLP and CV datasets.**
>
> **R:** Due to the relatively antiquated nature and the performance saturation on the STL-10 and AG News datasets, our approach did not achieve quite large gains on the average of multiple trials with different random seeds. But we can find out the fast-convergence and lightweight characteristics of our SemiReward. Consequently, we have supplemented our study with datasets from the CV and NLP domains with more difficult experimental settings (e.g., more class numbers and less labeled data) that exhibit remarkably superior performances.
>
> | NLP | PseudoLabel | PseudoLabel+**SR** | FlexMatch | FlexMatch+**SR** | SoftMatch | SoftMatch+**SR** | Gain |
> | --- | :---: | :---: | :---: | :---: | :---: | :---: | :---: |
> | amazon_review_250 | 53.45±1.90, | **49.13±0.77** | 45.73±1.60 | **43.08±0.11** | 45.29±0.95 | **42.98±0.24** | **3.09** |
> | amazon_review_1000 | 47.00±0.79 | **44.21±0.64** | 42.25±0.33 | **41.11±0.89** | 42.21±0.20 | **39.17±0.32** | **2.32** |
> | **CV** |  |  | |  |  |  |  |
> | semi_aves_3959_0 | 40.35±0.3 | **37.93±0.45** | 32.48±0.15 | **31.23±0.09** | 32.85±0.31 | **31.02±0.15** | **1.82** |
> | tissuemnist_80 | 56.92±4.54 | **53.06±0.11** | 58.36±3.8 | **54.27±0.71** | 58.24±3.08 | **53.52±1.07** | **4.22** |
>
> **Q1: Lack of explanation of the percentage of labels in the experimental setup.**
>
> **R:** The configuration of settings is guided by the approach outlined in the USB paper. Specific details and settings information can be found in Appendix Table A1. For your convenience in reference, we present them below:
>
> | Domain |    Dataset    | \#Label per class | \#Training data | \#Validation data | \#Test data | \#Class |
> |:------:|:-------------:|:-----------------:|:---------------:|:-----------------:|:-----------:|:-------:|
> |        |   CIFAR-100   |       2 / 4       |      50,000     |         -         |    10,000   |   100   |
> |   CV   |     STL-10    |       4 / 10      | 5,000 / 100,000 |         -         |    8,000    |    10   |
> |        |    EuroSat    |       2 / 4       |      16,200     |         -         |    5,400    |    10   |
> |        |    ImageNet   |        100        |     1,28,167    |         -         |    5,0000   |   1000  |
> |        |  Yelp Review  |      50 / 200     |     250,000     |       25,000      |    50,000   |    5    |
> |   NLP  |    AG News    |      10 / 50      |     100,000     |       10,000      |    7,600    |    4    |
> |        | Yahoo! Answer |      50 / 200     |     500,000     |       50,000      |    60,000   |    10   |
> |        |     ESC-50    |       5 / 10      |      1,200      |        400        |     400     |    50   |
> |  Audio |  UrbanSound8k |      10 / 40      |      7,079      |        816        |     837     |    10   |
> |        |  FSDnoisy18k  |       52-171      |  1,772 / 15,813 |         -         |     947     |    20   |

---

> ### Author Response · Authors · 2023-11-19
>
> Dear reviewer,
>
> We appreciate your valuable feedback. In our response, we supplemented the relevant experiments you mentioned and clarified the settings of the relevant data sets for our experiments. Several new sets of CV and NLP data sets illustrate the effectiveness of our gain. At the same time, the actual number of different labels has also been added. If you have more suggestions, please feel free to make them.
>
> As you may know, unlike previous years, this year's discussion period only lasts until November 22nd, and we are getting closer to that deadline. We hope to discuss this with you in the meantime and will be happy to provide more information based on your feedback or further questions.
>
> If you are satisfied with our response, please consider updating your score. If you need any clarification, please feel free to contact us.

---

> > ### Comment · Reviewer_uket · 2023-11-20
> >
> > Thank you for your detailed response. I carefully read the rebuttal, and concerns were addressed and included in the main paper. Thus, I'm raising my confidence score.

---

> > > ### Author Response · Authors · 2023-11-20
> > >
> > > We are glad that you are satisfied with our response, and sincerely appreciate your effort in reviewing our rebuttal. Since the rebuttal period is approaching the end, feel free to comment if you have any other questions. We are more than pleased to discuss with you and further improve the quality of the manuscript.
> > >
> > > Best regards,
> > >
> > > Authors

---

### Official Review · Reviewer_xKm4 · 2023-11-01

**Soundness:** 2 fair
**Presentation:** 3 good
**Contribution:** 2 fair
**Rating:** 6
**Confidence:** 4

**Summary:**

This work addressed the major challenge in semi-supervised learning, i.e. evaluating the quality of pseudo labels in anomaly detection. A rewarder network is introduced to predict good quality pseudo labels by training on real and generated samples. The proposed reward network is demonstrated to improve multiple semi-supervised learning baselines.

**Strengths:**

Strength:

1. Evaluating the quality of pseudo label is a real and major challenge for semi-supervised learning.

2. The experiment evaluation is extensive covering audio, nlp and computer vision tasks. The proposed rewarder also consistently improves the performance of multiple state-of-the-art semi-supervised learning baselines.

**Weaknesses:**

Weakness:

1. It is not clear why this rewarder design could be effective. If such a rewarder function is effective can one infer the labels for unlabelled data by simply choosing the label that maximise the rewards score given an input sample? In addition, one may also infer the labels for testing data in the similar way. If this is true, does it mean one can use directly use the rewarder function for inference?

2. According to Fig. 7, the semi-supervised training stage still requires a hard threshold to generate pseudo labels. The proposed method does not reduce the overall amount of hyperparameters to tune compared against existing SSL baselines.

3. It is unclear the definition of fake label. Does it mean the fake label must deviate from the ground-truth label? But the objective of Eq (7) seems to encourage the fake label to be consistent with ground-truth label. There is not enough clarifications here.

4. The current design seems to mix student model f_S with generator. Such a design may affect the training of student model.

**Questions:**

A more clear analysis of why the rewarder can predict the label quality and why the rewarder is not used for inference are encouraged.

More discussions on hyper-parameters and the definitions of fake label are recommended.

---

> ### Author Response · Authors · 2023-11-16
> **Response to Reviewer xKm4 PART 1**
>
> **W1:** **Lack of clarity on the validity of the rewarder design and inference strategy.**
>
> **R:** Thanks for your concerns, which are the most challenging problems we tackled with the Rewarder. As you have considered, directly predicting pseudo-labels with the algorithmic flow of the Rewarder will fall back to the basic pseudo-labeling algorithm, where the teacher model generates pseudo-labels and confident scores (can be regarded as the quality in Thresholding methods). It cannot provide a credible evaluation of generated pseudo-labels, which are coupled with the student training and affected by the confirmation bias. Therefore, our Rewarder serves as the pseudo-label selection method that evaluates the quality of the given pseudo-label of unlabeled data. As introduced in Sec. 3.1, our Rewarder is learned to compute the reward score, and we will select the relatively high-quality pseudo-labels with reward scores above the average in a mini-batch. We have shown the proposed reward score is calibrated for pseudo-label evaluation and is easy to fit by the lightweight Rewarder. Meanwhile, whether we can filter out high-quality depends on both the student & teacher models and the Rewarder. At the early stage of the student training, the generated pseudo-labels might not be reliable due to the weak student & teacher models. It will cost impractical computational overhead if you try to generate high-quality pseudo-labels by maximizing the reward score. When the student & teacher models are knowledgeable during the middle training period, the high-quality pseudo-labels will be discriminated by the Rewarder since it is online optimized. Moreover, you can refer to the General Response for the summary of our algorithm.
>
> **W2: Introduce new hyperparameters in rewarder filtering in semi-supervised training.**
>
> **R:** In the context of $\tau$, it is important to note that it does not represent a new hard threshold hyperparameter; rather, it serves as a scoring threshold marker for the Rewarder. Our inference process entails scoring multiple candidate pseudo-labels for a given sample using the Rewarder and subsequently filtering them. This involves scoring pseudo-label candidates collectively, computing individual rewards for each, and then calculating the average reward score. The average reward score, denoted by the symbol $\tau$, serves as the filtering threshold. During this process, pseudo-labels with scores below the average reward (i.e., the threshold $\tau$) are excluded. As this threshold is the mean of a set of pseudo-label candidate lists, it remains controllable and adaptable to changes in the training process. This adaptability implies that we do not introduce additional hyperparameters through hard threshold methods.

---

> ### Author Response · Authors · 2023-11-16
> **Response to Reviewer xKm4 PART 2**
>
> **W3: Ambiguity in the definition of fake labels.**
>
> **R:** Throughout the training process, please refer to the general response. The primary function of the Generator is to generate a **fake label** distribution, aiding the Rewarder in learning high-quality mapping relationships. Importantly, the goal of generating **fake labels** is not to produce a set of high-quality pseudo-labels for the Rewarder to learn. Such a learning distribution tends to exhibit **confirmation bias**, which is _**why the pseudo-label results directly generated by the student model are not employed**_. The purpose of learning this mapping, which decouples the training processes of the student model and the Rewarder, is to mitigate the aforementioned this **confirmation bias**. In finer detail, by inputting a pair of **fake label** and sample, the Rewarder assigns a score, and the training objective involves calculating the cosine similarity between the **fake label** of a data sample and its ground-truth label. Throughout the entire training process, in the initial stage, the Rewarder learns from the sample and its corresponding **fake label** as input. The learning objective is the difference between the true label of the sample and the **fake label** (reduction is applied here), normalized using cosine similarity (as depicted in Figure 4 of the main text, where the calibration curve explains the rationale behind using cosine similarity). The Generator generates a **fake label** distribution with random quality. These **fake labels** are not incorporated into the training process of the student model but serve to assist in the training of the Rewarder. This approach enables the Rewarder to comprehensively explore various scenarios of the score mapping, enhancing its ability to evaluate pseudo-labels effectively.
>
> **W4: Potential negative impact of mixing current designs with Student model with generator.**
>
> **R:** The proposed training pipeline effectively severs the interdependence of the student and rewarder training processes, a measure taken to mitigate confirmation bias. Through the generator generating multiple sets of **fake labels** with varying qualities and computing their respective scores, the rewarder initializes its mapping from pseudo-labels and unlabeled data to reward scores. In the first stage, these candidate training samples aid the Rewarder in its initial convergence, as elaborated in **W3 response** and the **general response**. Moving to the second stage, the rewarder necessitates a more diverse set of samples for further optimization, having already undergone substantial training on labeled samples. Once again, we employ samples and their corresponding pseudo-labels as input. However, in this stage, the learning objective shifts to the disparity between the sample's high-quality pseudo-labels (utilized as true labels) and the pseudo-labels generated by the generator. This approach facilitates a more extensive exploration of additional unlabeled samples to refine the rewarder's optimization.

---

> ### Author Response · Authors · 2023-11-19
>
> Dear reviewer,
>
> We appreciate your valuable feedback. In our response, we have tried our best to answer your questions about the Rewarder evaluation method and fake label. We have restated the problem through the general response and elaborated more specifically on the part where you gave us a weakness. If you have more questions, please inform us freely.
>
> As you may know, unlike previous years, this year's discussion period only lasts until November 22nd, and we are getting closer to that deadline. We hope to discuss this with you in the meantime and will be happy to provide more information based on your feedback or further questions.
>
> If you are satisfied with our response, please consider updating your score. If you need any clarification, please feel free to contact us.

---

> > ### Comment · Reviewer_xKm4 · 2023-11-21
> >
> > Thanks very much for the detailed clarifications. However, I still have the concern over the design of rewarder which takes image and label as inputs and outputs a normalized score. If such a rewarder can really quantify the quality of label, a simple way to infer the best label could be taking the gradient of $\partial R(x^u,y^u)/\partial y^u$ and use gradient ascent to estimate the $y^u$ that maximize $R(x^u,y^u)$. I do not see any high computation cost for such an inference. Basically, my major concern is, if the rewarder can well quantify the quality of pseudo label what prevents us from using the rewarder function to further refine the pseudo labels?

---

> ### Author Response · Authors · 2023-11-22
> **Response to Concerns of the Rewarder Design**
>
> Thanks for your insightful comment. We answer your concerns from two aspects, i.e., the necessity of decoupled two-stage training and the possibility of further optimizing pseudo labels with the Rewarder. We also added relevant discussion in Appendix B.7 of the latest revision for your reference.
>
> ### **Why is the rewarder not trained together with the student model?**
> Here, the first type of training process is to optimize the student and the Rewarder together without the Generator, where the teacher model generates candidate pseudo labels for the student and the Rewarder. We call the first type coupled training and the proposed two-stage training decoupled training. The following table shows the categories of the input, output results, and functions of the relevant models.
> There are two reasons for decoupling the training process of the student and the Rewarder. Firstly, the Rewarder requires diverse pseudo-labels as the training data to fit the ground truth reward scores rather than deterministic high-performance labels. Secondly, the student and the Rewarder might suffer from confirmation bias. Therefore, if we directly utilize pseudo labels generated by the teacher model as the couple training, the Rewarder might be unstable during the early training period and cannot learn more information in the second stage.
>
> | Model | Input | Output | Effects on Student Training | Effects on Rewarder Training |
> |---|:---:|:---:|:---:|:---:|
> | Student Model $f_S$ | $\Omega(x^l)$ | Predicted label | - | - |
> | Teacher Model $f_T$ | $\Omega(x^u)$ | Pseudo label | Generating candidate pseudo-labels | - |
> | Rewarder $\mathcal{R}$ | $f(x^u),y^u$ | Reward score | Selecting reliable pseudo-labels | Training objective |
> | Generator $\mathcal{G}$ | $f(x^u)$ | Fake label | - | Assist Rewarder training |
>
> We conduct comparison experiments of the two training processes on CIFAR-100 (400 labels) based on FlexMatch in the following table (report top-1 accuracy). When using the coupled training of the student and the Rewarder, FlexMatch+SR yields worse performance than the baseline (82.12 vs. 82.20). Meanwhile, the proposed two-stage training decouples the student and the Rewarder by the Generator (aiming to maximize the reward score) and achieves a great trade-off between performance gains and speedup. As shown in Figure A2(a), in coupled training, the selected pseudo-labels are occasionally of slightly higher quality, but are unstable and affected by the student model, while decoupled training stably produces high-quality pseudo-labels.
>
> | Method | FlexMatch+SR |
> |---|:---:|
> | Coupled Training | 82.12 ($\times 1.0$) |
> | +Gradient Ascent |  82.23 ($\times 1.2$) |
> | Decoupled Training | 83.11 ($\times$**2.2**) |
> | +Gradient Ascent | **83.25**($\times 1.7$) |
>
> ### **Why is SemiReward unable to maximize the reward score to generate better pseudo labels?**
> Currently, the reward score output by the rewarder only evaluates the pseudo label. Thus, we can obtain the pseudo labels with a gradient ascent trick (GA) as you suggested: Given a mini-batch of selected pseudo labels, we modify them towards more high-quality pseudo labels (or fake labels) by maximizing the reward scores with a step of gradient ascent in the inference process of the Rewarder. In the coupled training process, the Rewarder can directly affect the pseudo labels generated by the teacher model. However, in the decoupled training, the pseudo label and reward score have no gradient backpropagation relationship with each other, while fake labels generated by the Generator are related to the reward scores. Therefore, we add these gradient-ascent enhanced fake labels to the selected pseudo labels for the student training.
>
> We also implement the gradient ascent trick on CIFAR-100 in the table above. When using the coupled training, FlexMatch+SR (GA) can only obtain a limited performance gain and speedup over the baseline. Applying the gradient ascent to the decoupled training will yield a little performance gain with more extra computational costs and cause unstable training. As shown in Figure A2 (b), the quality of fake labels is relatively random and diverse, and it is difficult to steadily obtain high-quality labels through the gradient ascent and directly apply to the student training. Therefore, we chose the proposed decoupled training without GA as the final version.
>
> **In conclusion**, on the one hand, the first coupled training process is sub-optimal for SemiReward, and we avoid the existence of confirmation bias through decoupling method training. On the other hand, if we try to enhance the pseudo labels with the gradient ascent trick, we need a more complex loss design and violate the original intention of making a fast-convergence and plug-and-play method. The construction idea of SemiRewarder can be more widely used in other fields.
> Thanks again for your efforts and valuable feedback! We are looking forward to your further response.

---

> > ### Comment · Reviewer_xKm4 · 2023-11-22
> >
> > Thanks for the additional explanations and evaluations. I think the overall idea is novel despite some designs deserve more investigations. I would like to raise my score.

---

> > > ### Author Response · Authors · 2023-11-22
> > >
> > > Thanks for the score improvement. Your opinions have greatly improved the quality of our manuscripts. If you have more questions later, we look forward to your further discussion!

---

### Author Response · Authors · 2023-11-16
**General Response**

Dear Reviewers,

Greetings!

We appreciate the efforts and valuable feedback from three reviewers who contributed to improving the quality and writing of our manuscript. We are pleased that the reviewers have recognized the highlights of our work, such as addressing a significant challenge in semi-supervised learning by systematically modeling pseudo-label quality through the rewarder **(xKm4 & DH5L)**, the readability of the paper, the completeness and generality of the experiments **(all reviewers)**, as well as the algorithm's effectiveness and generalization capabilities **(uket & DH5L)**. Our manuscript has been updated (the $\textcolor{red}{red}$-colored portions indicate where we updated), and we will tackle the concerns of each reviewer point-by-point.

In response to the reviewers' suggestions, we have incorporated additional results comparing our algorithm on a large-scale dataset, specifically ImageNet. Furthermore, we have expanded our experiments to provide a more comprehensive illustration of the versatility of our algorithm. The pertinent findings have been detailed in the updated version of the manuscript.

We also have noticed that there seem to be specific questions regarding the process of our SemiReward algorithm. Therefore, we would like to provide further clarification as follows:

* **Composition of the SemiReward Model:** The SemiReward model consists of two components: the Rewarder and the Generator. The Rewarder serves as a scoring model, evaluating the quality of pseudo labels for a given set of unlabeled data. The Generator is used to generate fake labels with diversities, which are of relatively random quality. These fake labels are not included in the training process of the student model, which avoids confirmation bias. Instead, they assist in the Rewarder training to traverse various qualities of labels, enabling the Rewarder to assess label quality accurately. The design of the Rewarder is largely motivated by the fact that in semi-supervised problems, most data lack ground-truth annotations. In such cases, direct calculation using cosine similarity is not feasible. However, Rewarder can utilize combinations of existing samples + true labels to learn a mapping of pseudo-labels to scores through cosine similarity calculations, enabling the evaluation of pseudo-labels beyond the labeled dataset. This mapping is fixed and learnable, where high-quality pseudo-labels correspond to high scores, and low-quality pseudo-labels correspond to low scores, fundamentally measured by the cosine similarity distance between the pseudo-label and its corresponding true label.

* **Inference Process of SemiReward:** Following the widely used pseudo-labeling pipeline, the student model (or the teacher model in self-training) generates a set of candidate pseudo-labels. Using the learned Rewarder, reward scores for each candidate pseudo-label can be predicted, and we compute the mean of reward scores within each group as the threshold. Pseudo-labels with scores greater than the mean score are selected for training the student, while others are abandoned.

* **Training Objectives of SemiReward:** In scenarios where a sample possesses both a ground-truth label and a pseudo-label, the quality of pseudo-labels is effectively assessed through normalized cosine similarity, as elucidated in the calibration curve presented in Figure 4. This constitutes the primary objective of our Rewarder in supervised learning. Throughout the training regimen of the Rewarder, considering the diverse false labels generated by the Generator, corresponding ground truth labels are available, facilitating the calculation of ground truth reward scores, which serve as the target for the Rewarder. In terms of the training distribution objective, Rewarder is designed to evaluate a spectrum of pseudo-labels ranging from low to high quality. To achieve this, Generator is instrumental in supplementing such a pseudo-label distribution, enabling Rewarder to attain a heightened level of training proficiency.

Broadly Speaking, our SemiReward is intended to facilitate the generalization of the rewarding concept rather than ad-hoc design with task-specific priors in the data quality evaluation. It indicates that the proposed training and inference pipelines are not only designed for pseudo-label evaluation in semi-supervised tasks but also useful for label evaluation in other training processes to enhance dataset quality.

Thank you again for your continued attention and valuable insights. We are looking forwards to your further responses!

Best regards,

Authors

---

### Author Response · Authors · 2023-11-19

Dear reviewer,

We greatly appreciate your valuable feedback. In our response, we conduct additional experiments on different number of labels and NLP and CV datasets, and demonstrate its lower complexity from the further elaboration of Rewarder’s training and inference process. In addition, we further clarified some unclear statements in the paper. We have incorporated all changes into the revised manuscript for your consideration. We hope your concerns have been addressed.

As you may know, unlike previous years, the discussion period this year can only last until November 22, and we are gradually approaching this deadline. We would like to discuss this with you during this time and would be happy to provide more information based on your feedback or further questions.

If you are satisfied with our response, please consider updating your score. If you need any clarification, please feel free to contact us.

---

### Meta-Review · Area_Chair_cTvs · 2023-12-06

**Metareview:**

This work describes a framework for evaluating the quality of pseudo-labels for semi-supervised learning using a learned "rewarder network". It is an add-on module that can incorporated into existing semi-supervised learning methods. Extensive experiments on multiple datasets from CV, NLP and audio domains show improvements in accuracy and training speed. All reviewers thought the formulation was novel and appreciated the strong and extensive experiments, though there were some initial concerns about the formulation; these concerns were partially resolved through the discussion and all reviewers were positive about the paper after the discussion period. Based on the unanimous positive reviews, the AC recommends that the paper be accepted.

Nonetheless, the paper would benefit from a more detailed discussion about how the rewarder works as per the concerns of Reviewer xKm4 - in some sense it has to understand where the student model makes mistakes, and it is not immediately obvious how this will happen in the decoupled training scenario proposed.

**Justification For Why Not Higher Score:**

- Rewarder network and training scheme could use more justification

**Justification For Why Not Lower Score:**

- Novel formulation for semi-supervised learning
- Strong experimental results across multiple tasks and datasets

---

### Decision · Program_Chairs · 2024-01-16

Accept (poster)